# On the ion-inertial range density-power spectra in solar wind turbulence

**Rudolf A. Treumann**[1,3], **Wolfgang Baumjohann**[2], **and Yasuhito Narita**[2]

[1]International Space Science Institute, Bern, Switzerland
[2]Space Research Institute, Austrian Academy of Sciences, Graz, Austria
[3]Geophysics Department, Ludwig-Maximilians-University Munich, Germany,
*Correspondence to*: Wolfgang.Baumjohann@oeaw.ac.at

**Abstract.**

A model-independent first-principle first-order investigation of the shape of turbulent density-power spectra in the ion-inertial range of the solar wind at 1 AU is presented. De-magnetised ions in the ion-inertial range of quasi-neutral plasmas respond to Kolmogorov (K) or Iroshnikov-Kraichnan (IK) inertial-range velocity turbulence power spectra via the spectrum of the velocity-turbulence-related random-mean-square induction-electric field. Maintenance of electrical quasi-neutrality by the ions causes deformations in the power spectral density of the turbulent density fluctuations. Assuming inertial-range K (IK) spectra in solar wind velocity turbulence and referring to observations of density-power spectra suggests that the occasionally observed scale-limited bumps in the density-power spectrum may be traced back to the electric ion response. Magnetic power spectra react passively to the density spectrum by warranting pressure balance. This approach still neglects contribution of Hall currents and is restricted to the ion-inertial range scale. While both density and magnetic turbulence spectra in the affected range of ion-inertial scales deviate from K or IK power law shapes, the velocity turbulence preserves its inertial-range shape in this process to which spectral advection turns out to be secondary but may become observable under special external conditions. One such case observed by WIND is analysed. We discuss various aspects of this effect including the affected wavenumber scale range, dependence on angle between mean flow velocity and wavenumbers and, for a radially expanding solar wind flow, assuming adiabatic expansion at fast solar wind speeds and a Parker dependence of the solar wind magnetic field on radius, also the presumable limitations on the radial location of the turbulent source region.

**Keywords.** Solar wind, turbulence, inertial range power spectrum, ion-inertial range, density power spectrum

## 1 Introduction

The solar wind is a turbulent flow of origin in the solar corona. It is believed to become accelerated within a few solar radii in the coronal low-beta region. Though this awaits approval, it is also believed that its turbulence originates there. Turbulent power spectral densities in the solar wind have been measured *in situ* around 1 AU for several decades already. They include spectra of the magnetic field (cf. e.g., Goldstein et al., 1995; Tu & Marsch, 1995; Zhou et al., 2004; Podesta, 2011, for reviews among others), but with improved instrumentation also of the fluid velocity (Podesta et al., 2007; Podesta, 2009; Šafránková et al., 2013), electric field (Chen et al., 2011, 2012, 2014a, b), temperature (Šafránková et al., 2016) and (starting with Celnikier et al., 1983, who already reported its main properties) also of the (quasi-neutral) solar wind density (Chen et al., 2012; Šafránková et al., 2013, 2015, 2016).

Complementary to the measurements *in situ* the solar wind, ground-based observations of radio scintillations from distant stars, originally applied (Lee & Jokipii, 1975, 1976; Cordes et al. , 1991; Armstrong et al., 1995) to the interstellar medium (ISM) (for early reviews cf., e.g., Coles, 1978; Armstrong et al., 1981) and used for extra-heliospheric plasma diagnosis (cf., Haverkorn & Spangler, 2013) also provided information about the solar wind density turbulence (Coles & Harmon, 1989; Armstrong et al., 1990; Spangler & Sakurai, 1995; Harmon & Coles, 2005) mostly at solar radial distances $< 60 R_\odot \approx 0.25$ AU in the innermost solar wind very low $0.1 < \beta_i < 1$ (cf., e.g., the model of McKenzie et al.,

1995) region, which is of particular interest because it is the presumable source region of the solar wind being accessible only from remote. Solar wind turbulence generated here seems frozen to,[1] and afterwards being transported radially outward by the flow. Radio-phase scintillation of spacecraft signals from Viking, Helios and Pioneer have been used early on (Woo & Armstrong, 1979) to determine solar wind density power spectra in the radial interval $\leq 1$ AU reporting mean spectral Kolmogorov slopes $\sim -\frac{5}{3}$ with a strong flattening of the spectrum near the Sun at distances $< 30 R_\odot$ where the slope flattens down to $\sim -\frac{7}{6} = -1.1$, a finding which suggests evolution of the density turbulence with solar distance. In the ISM radio scintillation observations covered a huge range of decades from wavelength scales $\lambda \approx 15$ AU down to close to the Debye length $\lambda_D \approx 50$ m, suggesting an approximate Kolmogorov spectrum over 7 decades. From recent *in situ* Voyager 1 observations of ISM electron densities (Gurnett et al., 2013) a Kolmogorov spectrum has been inferred down to wavelengths of $\lambda \sim 10^6$ m that is followed by an adjacent spectral intensity excess on the assumed kinetic scales for wave lengths $\lambda \gtrsim \lambda_D$ (Lee & Lee, 2019)

Density fluctuations $\delta N$ are generally inherent to pressure fluctuations $\delta P$. From fundamental physical principles it follows that density turbulence does not evolve by itself. Through the continuity equation it is related to velocity turbulence, which in its course requires the presence of free energy, being driven by external forces. It is primary while turbulence in density, temperature, and magnetic field is secondary. Density turbulence may signal the presence of a population of compressive (magnetoacoustic-like) fluctuations in addition to the usually assumed (cf., e.g., Biskamp, 2003; Howes, 2015) alfvénic turbulence, the dominant fluid-magnetic fluctuation family dealing with the mutually related alfvénic velocity and magnetic fields made use of in MHD theory based on Elsasser variables (Elsasser, 1950).

Inertial range velocity turbulence is subject to Kolmogorov (Kolmogorov, 1941a, b, 1962) or Iroshnikov-Kraichnan (Iroshnikov, 1964; Kraichnan, 1965, 1966, 1967) turbulence spectra respectively their generalisation to anisotropy with respect to any mean magnetic field (Goldreich & Sridhar, 1995). In the solar wind, Kolmogorov inertial range spectra reaching down into the presumable dissipation range have been confirmed by a wealth of *in situ* observations (cf., e.g., Goldstein et al., 1995; Tu & Marsch, 1995; Zhou et al., 2004; Alexandrova et al., 2009; Boldyrev et al., 2011;

---

[1] We do not touch on the subtle question whether in a low-beta/strong-field plasma any frozen turbulence on MHD-scales above the ion cyclotron radius can evolve. According to inferred spatial anisotropies, it seems that close to the Sun turbulence in the density is almost field-aligned. On the other hand, ion-inertial range turbulence at shorter scales will be much less affected. It can be considered to be isotropic. Near 1 AU, where most *in situ* observations take place, one has $\beta \gtrsim 1$. One may expect that turbulence here also contains contributions which are generated locally, if only some free energy would become available.

Matthaeus et al., 2016; Lugones et al., 2016; Podesta, 2011; Podesta et al., 2006, 2007; Sahraoui et al., 2009, and others). Since the mean fields $B_0, T_0, N_0, U_0$ themselves obey pressure balance, one has for pressure balance among the turbulent fluctuations

$$\frac{\langle|\delta \boldsymbol{B}|^2\rangle}{B_0^2} = \frac{\sqrt{\langle|\delta N|^2\rangle}}{N_0} + \frac{\sqrt{\langle|\delta T|^2\rangle}}{T_0} \tag{1}$$

The angular brackets $\langle\ldots\rangle$ indicate averaging over the spatial scales of the turbulence respectively turbulent fluctuations. Alfvénic fluctuations (cf., e.g., Howes, 2015, for a recent theoretical account of their importance in MHD turbulence) compensate separately due to their magnetic and velocity fluctuations being related; they do not contribute to extra compression. In order to infer the contribution of density fluctuations, one compares their spectral densities with those of the temperature $\delta T$ or magnetic field $\delta \boldsymbol{B}$. This requires normalisation to the means. Solar wind densities at 1 AU are of the order of $N_0 \sim 10$ cm$^{-3}$, while ion thermal speeds are of the order of $v_i \sim 30$ km s$^{-1}$. Moreover, mean plasma betas are of order $\beta_i \sim 1$ here. For checking pressure balance, measured density fluctuations can be compared with those two.

An example is shown in Fig. 1 based on solar wind measurements on July 6, 2012 (Šafránková et al., 2015, 2016). There is not much freedom left in choosing the mean densities and temperatures in Fig. 1. Densities at 1 AU barely exceed 10 cm$^{-3}$. Electron temperatures are insensitive to those low frequency density fluctuations. High mobility makes electron reaction isothermal.

The data in Figure 1 show the relative dominance of density fluctuations over ion temperature fluctuations under moderately-low speed solar wind conditions at all frequencies larger than the lowest accessible MHD frequencies. This is not surprising because one would not expect large temperature effects. Ion heating is a slow process which does not react to any fast pressure fluctuations caused by density or magnetic turbulence. It just shows that the turbulent thermal pressure is mainly due to density fluctuations over most of the frequency range. In the low frequency MHD range the kinetic pressure of large-scale turbulent eddies dominates.

Inertial range power spectra of turbulent density fluctuations are power laws. Occasionally they exhibit pronounced spectral excursions from their monotonic course prior to drop into the dissipative range. Whenever this happens, the spectrum flattens or, in a narrow range of scales, even turns to positive slopes, sometimes dubbed spectral "bumps". The reason for such spectral excesses still remains unclear. Similar bumps have also been seen in electric field spectra (cf., e.g., Chen et al., 2012) where they have tentatively been suggested to indicate the presence of kinetic Alfvén waves which may be excited in the Hall-MHD (cf., e.g., Huba, 2003) range as eigenmodes of the plasma. Models including Alfvén ion-cyclotron waves (Harmon & Coles, 2005), or kinetic Alfvén waves (Chandran et al., 2009) have been proposed to cause

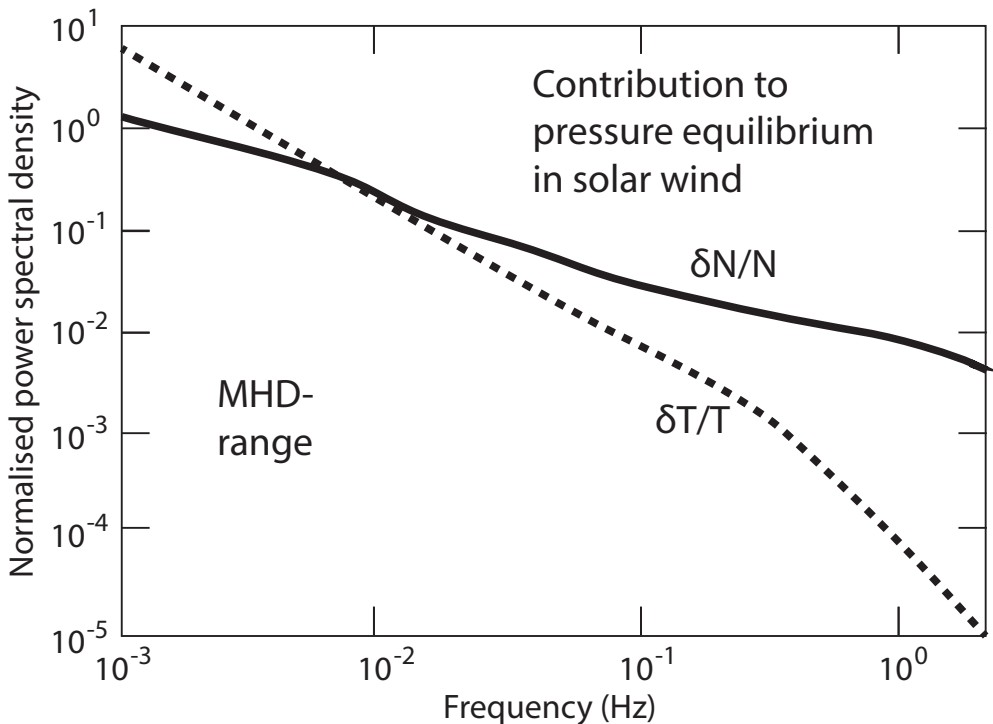

**Figure 1.** Normalised solar wind power spectra of turbulent temperature and density fluctuations. The curves are based on data from Šafránková et al. (2016) obtained on July 6, 2012 from the Bright Monitor of the Solar Wind (BMSW) instrument on board the Spektr-R spacecraft. The solar wind conditions of these observations have been tabulated (Chen et al., 2014a). They indicate rather slow than medium conditions. The data have been rescaled and normalised to the main density $N_0$ and temperature $T_0$ in order to show their relative contributions to an assumed solar wind pressure balance. The interesting result is that in the lowest MHD frequency range density fluctuations are irrelevant with respect to pressure balance. At higher frequencies, however, the density fluctuations dominate the temperature fluctuations.

spectral flattening. Kinetic Alfvén waves may also lead to bumps if only $\beta_i \ll 1$. In fact, kinetic Alfvén waves possess large perpendicular wave number $k_\perp \lambda_i \sim 1$ of the order of the inverse ion-inertial length (cf., e.g., Baumjohann & Treumann, 1996), the scale on that ions demagnetise. If sufficient free energy is available, they can thus be excited and propagate in this regime (cf., e.g., Gary, 1993; Treumann & Baumjohann, 1997). Recently Wu et al. (2019) provided kinetic-theoretical arguments for kinetic Alfvén waves contributing to turbulent dissipation in the ion-inertial scale region. Causing bumps, the waves should develop large amplitudes on the background of general turbulence, i.e. causing intermittency. This requires the presence of a substantial amount of unidentified free energy, for instance in the form of intense plasma beams, which are very well known in relation to collisionless shocks both upstream and downstream (cf., e.g., Balogh & Treumann, 2013). If kinetic Alfvén waves are unambiguously confirmed, the inner solar wind at $\lesssim 0.6$ AU could be subject to the continuous presence of small scale collisionless shocks, a not unreasonable assumption which would be supported by observation of sporadic nonthermal

coronal radio emissions (type I through type IV solar radio bursts).

In the present note we take a completely different *model independent* point of view avoiding reference to any superimposed plasma instabilities or intermittency (e.g., Chen et al., 2014b). We do not develop any "new theory" of turbulence. Instead, we remain in the realm of turbulent fluctuations, asking for the effect of ion inertia, respectively ion-demagnetisation in the ion-inertial Hall-MHD range, on the shape of the inertial-range power spectral density which will be illustrated referring to a few selected observations. To demonstrate pressure balance we refer to related magnetic power spectra, both measured *in situ* aboard spacecraft which require a rather sophisticated instrumentation. Those measurements were anticipated by indirectly inferred density spectra in the solar wind (Woo, 1981; Coles & Filice, 1985; Bourgeois et al., 1985) and the interstellar plasma (Coles, 1978; Armstrong et al., 1981, 1990) from detection of ground based radio scintillations.

In the next section we discuss the response of demagnetised ions to the presence of turbulence on scales between the ion and electron inertial lengths. This response we interpret

as the consequence of electric field fluctuations in relation to the turbulent velocity field. Requirement of charge neutrality maps them to the density field via Poisson's equation. The additional contribution of the Hall effect can be separated. We then refer to turbulence theory, assuming that the mechanical inertial-range velocity turbulence spectrum is either Kolmogorov (K) or Iroshnikov-Kraichnan (IK) and, in a fast streaming solar wind under relatively weak conditions (Treumann et al., 2019), maps from wavenumber $k$ into a stationary observer's frequency $\omega_s$ space via Taylor's hypothesis (Taylor, 1938).

In order to be more general, we split the mean flow velocity into bulk $V_0$ and large eddy $U_0$ velocities, the latter known (Tennekes, 1975) to cause Doppler broadening of the local velocity spectrum at fixed wavenumber (reviewed and backed by numerical simulations by Fung et al., 1992; Kaneda, 1993). Imposing the theoretical K or IK inertial range spectra, we then find the deformed power density spectra of density turbulence versus spacecraft frequency. We apply these to some observed spectral density bumps which we check on a measured magnetic power spectrum for pressure balance. The results are tabulated. Since bumpy spectra are rather rare, we also consider two more "normal" bumpless though deformed density-power spectra which exhibit some typical spectral flattening and were obtained under different solar wind conditions. The letter concludes with a brief discussion of the results.

## 2   Inertial range ion response

Our main question concerns the cause of the occasionally observed scale-limited bumps in the turbulent density power spectra, in particular their deviation from the expected monotonic inertial range power law decay towards high wavenumbers prior to entering the presumable dissipation range.

The philosophy of our approach is the following. Turbulence is always mechanical, i.e. in the velocity. It obeys a turbulent spectrum which extends over all scales of the turbulence. In a plasma, containing charged particles of different mass, these scales for the particles divide into magnetised, inertial, unmagnetised, and dissipative. On each of these intervals, the particles behave differently, reacting to the turbulence in the velocity. In the inertial range, the particles lose their magnetic property. They do not react to the magnetic field. They, however, are sensitive to the presence of electric fields, independent of their origin. Turbulence in velocity in a conducting medium in the presence of external magnetic fields is always accompanied by turbulence in the electric field due to gauge invariance, respectively the Lorentz force. This electric field affects the unmagnetised component of the plasma, the ions in our case, which to maintain quasi-neutrality tend to compensate it. Below we deal with this effect and its consequences for the density power-density spectrum.

### 2.1   Electric field fluctuations in the ion-inertial range

The steep decay of the normalised fluctuations in ion temperature above frequencies $> 10^{-1}$ Hz is certainly due to the drop in ion dynamics at frequencies close to and exceeding the ion cyclotron frequency, which at 1 AU distance from the sun is of the order of $f_{ci} = \omega_{ci}/2\pi \sim 1$ Hz for a nominal magnetic field of $\sim 10$ nT. In this range we enter the (dissipationless) ion inertial or Hall (electron-MHD) domain where ions demagnetise, currents are carried by magnetised electrons, both species decouple magnetically and Hall currents arise. At those frequencies, far below the electron $f_e = \omega_e/2\pi \sim 35$ kHz and (assuming protons) ion $f_i = \omega_i/2\pi \sim 0.8$ kHz plasma frequencies, ions and electrons couple mainly through the condition of quasi-neutrality, i.e. via the turbulent induction electric field which becomes[2]

$$
\begin{aligned}
\delta \boldsymbol{E} = & - & \delta \boldsymbol{V}_\perp \times \boldsymbol{B}_0 - (\boldsymbol{V}_0 + \boldsymbol{U}_0) \times \delta \boldsymbol{B} - \\
& - & \frac{1}{eN_0} \boldsymbol{B}_0 \times \delta \boldsymbol{J} + \\
& + & \left\langle \delta \boldsymbol{V} \times \delta \boldsymbol{B} + \frac{1}{eN_0} \delta \boldsymbol{J} \times \left( \frac{\delta N}{N_0} \boldsymbol{B}_0 - \delta \boldsymbol{B} \right) \right\rangle
\end{aligned}
\tag{2}
$$

For later use, we split the main velocity field $\langle \boldsymbol{V} \rangle = \boldsymbol{V}_0 + \boldsymbol{U}_0$ into the bulk flow (convection) $\boldsymbol{V}_0$ and an advection velocity $\boldsymbol{U}_0$. The latter is the mean velocity of a small number of large eddies which carry the main energy of the turbulence. Even for stationary turbulence they advect the bulk of small-scale eddies around at speed $\boldsymbol{U}_0$ (Tennekes, 1975; Fung et al., 1992).

The last three averaged nonlinear terms within the angular brackets $\langle \dots \rangle$ on the right are the nonlinear contributions of the fluctuations to the mean fields yielding an electromotive force which contributes to mean field processes like convection, dynamo action, and turbulent diffusion. They vary only on the large mean-field scale. On the fluctuation scale they are constant and can be dropped, unless the turbulence is bounded, in which case boundary effects must be taken into account at the large scales of the system. Generally, in the solar wind this is not the case. The remaining three linear terms distinguish between directions parallel and perpendicular to the main magnetic field $\boldsymbol{B}_0$. The third linear term being the genuine perpendicular Hall contribution. From Ampere's law for the current fluctuation $\mu_0 \delta \boldsymbol{J} = \nabla \times \delta \boldsymbol{B}$ we have for the perpendicular and parallel components of the turbulent elec-

---

[2]This equation is easily obtained by standard methods when splitting the fields in the ideal (collisionless) Hall-MHD Ohm's law $\boldsymbol{E} = -\boldsymbol{V} \times \boldsymbol{B} + (1/eN) \boldsymbol{J} \times \boldsymbol{B}$ with $\boldsymbol{E}, \boldsymbol{B}, \boldsymbol{V}, \boldsymbol{J}$ electric, magnetic, and current fields, respectively, into mean (index 0) and fluctuating fields according to $\boldsymbol{E} = \boldsymbol{E}_0 + \delta \boldsymbol{E}$ etc.; averaging over the fluctuation scales, with $\langle \dots \rangle$ indicating the averaging procedure, yields the mean field electric field equation. Subtracting it from the original equation produces the wanted expression of the turbulent electric fluctuations $\delta \boldsymbol{E}$ through the mean and fluctuating velocity and magnetic fields.

tric field

$$
\begin{aligned}
\delta \boldsymbol{E}_\perp &= \boldsymbol{B}_0 \times \left[ \delta \boldsymbol{V}_\perp - \frac{U_{0\parallel}}{B_0} \delta \boldsymbol{B}_\perp - \right. \\
&\quad \left. - \frac{1}{e\mu_0 N_0} \left( \nabla \times \delta \boldsymbol{B} \right)_\perp \right] - \boldsymbol{U}_{0\perp} \times \delta \boldsymbol{B}_\parallel \qquad (3) \\
\delta \boldsymbol{E}_\parallel &= -\boldsymbol{U}_{0\perp} \times \delta \boldsymbol{B}_\perp \implies 0
\end{aligned}
$$

5 The second of these equations is of no interest, because the low frequency parallel electric field its right-hand side produces is readily compensated by electron displacements along $\boldsymbol{B}_0$.

This leaves us with the fluctuating perpendicular induc-10 tion field in the first Eq. (3). Here, any parallel advection $U_{0\parallel}$ attributes to the perpendicular velocity fluctuations from perpendicular magnetic fluctuations $\delta \boldsymbol{B}_\perp$. On the other hand, any present parallel compressive magnetic fluctuations $\delta \boldsymbol{B}_\parallel = \boldsymbol{B}_0 (\delta B_\parallel / B_0)$ add through perpendicular advection 15 $\boldsymbol{U}_{0\perp}$. In their absence, when the magnetic field is non-compressive, the last term disappears.

The complete Hall contribution to the electric field, viz. the last term in the brackets in Eq. (3), can be written as

$$
\delta \boldsymbol{E}_\perp^H = -\frac{B_0}{e\mu_0 N_0} \left( \boldsymbol{\nabla}_\perp \delta B_\parallel - \nabla_\parallel \delta \boldsymbol{B}_\perp \right) \qquad (4)
$$

20 Even for $U_{0\parallel} = 0$ it contributes through the turbulent fluctuations in the magnetic field. As both these contributions depend only on $\delta \boldsymbol{B}$ we can isolate them for separate consideration. One observes that, in the absence of any compressive magnetic components $\delta B_\parallel$ and homogeneity along the mean 25 field $\nabla_\parallel = 0$, there is no contribution of the turbulent Hall term to the electric induction field. In that case only velocity turbulence contributes. Below we consider this important case.

## 2.2 Relation to density fluctuations: Poisson's equation

30 Let us assume that advection by large-scale energy-carrying eddies is perpendicular $\boldsymbol{U}_0 = \boldsymbol{U}_{0\perp}$, and there are no compressive magnetic fluctuations $\delta \boldsymbol{B}_\parallel = 0$. In Eq. (2) this reduces to considering only the first term containing the velocity fluctuations. We ask for its effect on the density fluc-35 tuations in the ion-inertial domain on scales where the ions demagnetise.

On scales in the ion-inertial range shorter than either the ion thermal gyroradius $\rho_i = v_i / \omega_{ci}$ or – depending on the direction to the mean magnetic field $\boldsymbol{B}_0$ and the value of 40 plasma beta $\beta = 2\mu_0 N_0 T_0 / B_0^2$, with $\omega_{ci} = eB_0/m_i$ ion cyclotron and $\omega_i = e\sqrt{N_0/\epsilon_0 m_i}$ ion plasma frequency, respectively – inertial length $\lambda_i = c/\omega_i$, the ions demagnetise. Being non-magnetic, they do not distinguish between potential and induction electric fields. They experience the induction 45 field caused by the spectrum of velocity fluctuations as an external electric field which, in an electron-proton plasma, causes a charge density fluctuation $e\delta N_i = e\delta N_e$ and thus a

density fluctuation $\delta N$. Poisson's equation implies that

$$
\nabla \cdot \delta \boldsymbol{E} = \frac{e}{\epsilon_0} \delta N \qquad \implies \qquad i\boldsymbol{k} \cdot \delta \boldsymbol{E}_{\boldsymbol{k}} = \frac{e}{\epsilon_0} \delta N_{\boldsymbol{k}} \quad (5)
$$

The right expression is its Fourier transform. For complete-50 ness we note that the Hall contribution to the Poisson equation in Fourier space reads

$$
i\boldsymbol{k}_\perp \cdot \delta \boldsymbol{E}_{\perp \boldsymbol{k}}^H = \frac{B_0}{e\mu_0 N_0} \left( k_\perp^2 \delta B_{\parallel \boldsymbol{k}} - k_\parallel \boldsymbol{k}_\perp \cdot \delta \boldsymbol{B}_{\perp \boldsymbol{k}} \right) = \frac{e}{\epsilon_0} \delta N_{\boldsymbol{k}}^H
$$
$$
(6)
$$

Again it becomes obvious that absence of parallel (compressive) magnetic turbulence eliminates the first term in 55 this expression while purely perpendicular propagation eliminates the second term. Alfvénic turbulence, for instance, with $\delta B_\parallel = 0$ and $k_\perp = 0$ has no Hall-effect on the modulation of the density spectrum, a fact which is well known. On the other hand, for perpendicular wavenumbers $k = k_\perp$ only 60 compressive Hall-magnetic fluctuations $\delta B_{\parallel k_\perp}$ contribute to the Hall-fluctuations in the density $\delta N_{k_\perp}^H$.

## 2.3 Relation between density and velocity power spectra

We are interested in the power spectrum of the turbulent den-65 sity fluctuations in the proper frame of the turbulence.

Multiplication of the only remaining first term in the electric induction field Eq. (3) with wavenumber $\boldsymbol{k}$ selects wavenumbers $\boldsymbol{k}_\perp$ perpendicular to $\boldsymbol{B}_0$. Combination of Eq. (2) and the Poisson equation then yields an expression for 70 the power spectrum of the turbulent density fluctuations[3] in wavenumber space

$$
\langle |\delta N|^2 \rangle_{k_\perp} = \left( \frac{\epsilon_0 B_0}{e} \right)^2 k_\perp^2 \langle |\delta \boldsymbol{V}|^2 \rangle_{k_\perp} \qquad (7)
$$

where we from now on drop the index $\perp$ on the velocity $\delta \boldsymbol{V}_\perp$. Angular brackets again symbolise spatial averag-75 ing over the fluctuation scale. The functional dependence on wavenumber is indicated by the index $k_\perp$. It is obvious that the power spectrum of density fluctuations in the ion-inertial Hall MHD domain is completely determined by the power spectrum of the turbulent velocity.[4] This can be written as 80

$$
\begin{aligned}
\frac{\langle |\delta N|^2 \rangle_{k_\perp}}{N_0^2} &= \left( \frac{V_A}{c} \right)^2 \left( \frac{k_\perp}{\omega_i} \right)^2 \langle |\delta \boldsymbol{V}|^2 \rangle_{k_\perp} \\
&= \frac{\langle |\delta \boldsymbol{V}|^2 \rangle_{k_\perp}}{c^2} \left( \frac{V_A}{c} \right)^2 \left( k_\perp \lambda_i \right)^2
\end{aligned} \qquad (8)
$$

---

[3] The procedure of obtaining the power spectrum is standard. So we skip the formal steps which lead to this expression.

[4] One may object that, at smaller wavenumbers outside the ion-inertial range, this would also be the case, which is true. There reference to the continuity equation, for advection speeds $U_0 \neq 0$, yields $\langle |\delta N|^2 \rangle_{\boldsymbol{k}} = N_0^2 \langle |\boldsymbol{k} \cdot \delta \boldsymbol{V}|^2 \rangle_{\boldsymbol{k}} / (\boldsymbol{k} \cdot \boldsymbol{U}_0)^2$, which is obtained without reference to Poisson's equation. However, its dependence on wave number is different and, in addition, it is undefined for vanishing advection. In the absence of advection the density spectrum is determined from the equation of motion by simple pressure balance.

where $V_A^2 = B_0^2/\mu_0 m_i N_0$ is the squared Alfvén speed, and $\omega_i^2 = e^2 N_0/\epsilon_0 m_i$ is the squared proton plasma frequency. As expected, in order to contribute to density fluctuations, perpendicular scales $\lambda_\perp < \lambda_i$ smaller than the ion inertial length $\lambda_i = c/\omega_i$ are required, while in the long-wavelength range $k_\perp \lambda_i < 1$ there is no effect on the spectrum. This is in agreement with the assumption that any spectral modification is expected only in the ion inertial range.

The last equation is the main formal result. It is the wanted relation between the power spectra of density and velocity fluctuations. It contains the response of the unmagnetised ions to the mechanical turbulence.

## 2.4 Affected scale range

The density response demand that the ions are unmagnetised. This implies that $k_\perp \rho_i < 1$, where $\rho_i = v_i/\omega_{ci} = \lambda_i v_i/V_A$ is the ion gyroradius, with $v_i$ the thermal speed. Thus we have two conditions which must simultaneously be satisfied

$$k_\perp \lambda_i > 1 \quad \text{and} \quad k_\perp \lambda_i > \frac{V_A}{v_i} \equiv \beta_i^{-\frac{1}{2}} \qquad (9)$$

For $V_A < v_i$ the second condition is trivial. This is, however, a rare case. So the more realistic restriction is the opposite small ion-beta case when $V_A > v_i$ and hence $\beta_i < 1$. It must, however, be combined with another condition which requires that the wavenumbers should be smaller than the inverse electron gyroradius $\rho_e = v_e/\omega_{ce}$. The relation between $\rho_e$ and $\rho_i$ is $\rho_e^2/\rho_i^2 = m_e T_e/m_i T_i$. Moreover we have $\rho_i/\lambda_i = v_i/V_A$ and in addition $\beta_i = v_i^2/V_A^2 = (T_i/T_e)\beta_e$. Using all these relations we obtain finally that

$$1 < k_\perp^2 \lambda_i^2 \beta_i < \frac{m_i}{m_e} \frac{T_i}{T_e} \quad \text{for} \quad \beta_i < 1 \qquad (10)$$

This expression defines the marginal condition for the existence of a range in wavenumbers where the ions respond to the spectrum of the turbulent electric field $\delta\boldsymbol{E}$

$$\frac{T_i}{T_e} \gtrsim \frac{m_e}{m_i} \sim 0.001 \qquad (11)$$

Because of the smallness of the right-hand side this is a weak restriction. As expected, any effect on the density power spectrum will disappear at wavenumbers $k_\perp \rho_e$ where the electrons demagnetise. On the other hand, the lower wavenumber limit is a sensitive function of the external conditions. This becomes clear when writing it in the form

$$T_e/T_i \beta_e < k_\perp^2 \lambda_i^2 \qquad (12)$$

The electron plasma beta in the solar wind is of order $\beta_e \gtrsim O(1)$. However, the temperature ratio $T_e/T_i$ is variable and usually large, varying between a few and a few tens. Thus usually $\beta_i < 1$. Figure 2 shows a graph of this dependence.

## 2.5 Application to K and IK inertial range models of turbulence

The power spectrum of the Poisson-modified ion-inertial range density turbulence can be inferred, once the power spectral density of the velocity is given. This spectrum must either be known a priori or requires reference to some model of turbulence.

Ourselves, we do not develop any model of turbulence here. In application to the solar wind we just make, in the following, use of the Kolmogorov (K) spectrum (or its anisotropic extension by Goldreich & Sridhar, 1995, abbreviated KGS) but will also refer to the Iroshnikov-Kraichnan (IK) spectrum which both have previously been found to be of relevance in solar wind turbulence.

We shall make use of those spectra in two forms: the original ones which just assume stationarity and absence of any bulk flows, and their modified advected extensions. The latter account for a distinction between a small number of large energy-carrying eddies with mean eddy vortex speed $\boldsymbol{U}_0$ and bulk turbulence consisting of large numbers of small energy-poor eddies which are frozen to the large eddies. The large eddies stir the small-scale turbulence forcing it into *advective* motion (Tennekes, 1975). This causes Doppler broadening of the wavenumber spectrum at fixed $k$ and has been confirmed by numerical simulations (Fung et al., 1992; Kaneda, 1993). Below, it will be found that this advection cannot be resolved in bulk convective flow which buries the subtle effect of Doppler broadening. A probable counter example is shown in Fig. 5.

The stationary velocity spectrum of turbulent eddies at energy injection rate $\epsilon$ exhibits a broad inertial power law range in $\boldsymbol{k}$ (Kolmogorov, 1941a, b, 1962; Obukhov, 1941) which, between injection $k_{in}$ and dissipation at $k_d$ wavenumbers, obeys the famous isotropic Kolmogorov power spectral density law in wavenumber space

$$\langle |\delta\boldsymbol{V}|^2 \rangle_k \equiv \mathcal{E}_K(k) = C_K \epsilon^{\frac{2}{3}} k^{-\frac{5}{3}} \qquad \text{for} \qquad k_{in} < k < k_d \qquad (13)$$

with $C_K \approx 1.65$ Kolmogorov's constant of proportionality (as determined by Gotoh & Fukayama, 2001, using numerical simulations). Clearly, when in a fast streaming solar wind straightforwardly mapping this K spectrum by the Taylor hypothesis (Taylor, 1938) into the stationary spacecraft frame, the spectral index is unchanged, and one trivially recovers the $\omega_s^{-\frac{5}{3}}$ Kolmogorov slope in frequency space.

This changes drastically, when referring to an advected K spectrum of velocity turbulence (Fung et al., 1992; Kaneda, 1993) which yields the above mentioned spectral Doppler broadening at fixed $k$

$$\mathcal{E}_{k\omega_k}^{ad} = \frac{1}{2} \frac{\mathcal{E}_K(k)}{\sqrt{2\pi} k U_0} \sum_{\pm} \exp\left[ -\frac{1}{2} \frac{\omega_\pm^2}{(kU_0)^2} \right], \qquad (14)$$

$$\omega_\pm = \omega_k \pm \ell_K k^{\frac{2}{3}}$$

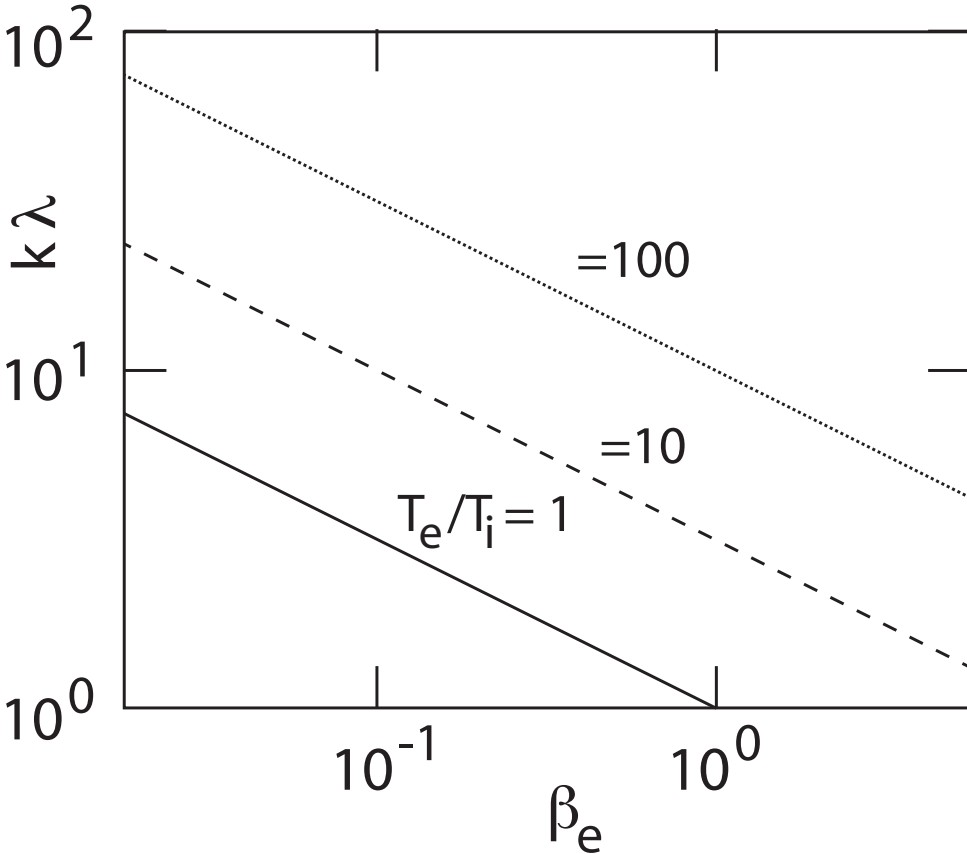

**Figure 2.** The range of permitted values of $k_\perp \lambda_i$ as function of $\beta_e$ for different ratios $T_e/T_i$. Only the range above the lines is relevant. In the solar wind usually $T_e > T_i$ implying $\beta_e > \beta_i$ (Newbury et al., 1998; Wilson III et al., 2018) unless the electrons become cooled by some process like emitting radiation, electron hole formation, or charge exchange.

which is due to decorrelation of the small eddies in advective transport, with $\ell_K \sim O(1)$ some constant. The $k^{\frac{2}{3}}$ dependence in the argument of the exponential results from advection $\boldsymbol{k} \cdot \delta\boldsymbol{V}$ of neighbouring eddies at velocity of $\delta V \propto k^{-\frac{1}{3}}$ (Tennekes, 1975; Fung et al., 1992). The frequency $\omega_k$ stands for the internal dependence of the turbulent frequency on the turbulent wavenumber $\boldsymbol{k}$. It can be understood as an internal "turbulent dispersion relation" which in turbulence theory is neglected.[5] Then the advected power spectrum at large $k$ is

power law

$$\mathcal{E}^{ad}_{k\omega_k} \quad \propto \quad \frac{\mathcal{E}_K(k)}{k} \exp\left(-\tfrac{1}{2}\cos^2\gamma_k\right) \sim k^{-\frac{8}{3}}, \tag{15}$$

$$\omega_\pm \quad \approx \quad \boldsymbol{k} \cdot \boldsymbol{U}_0 = kU_0\cos\gamma_k \tag{16}$$

In the stationary turbulence frame the power spectrum of turbulence in the velocity decays $\propto k^{-\frac{8}{3}}$ with non-Kolmogorov spectral index $\frac{8}{3} \approx 2.7$.

It is of particular interest to note that solar wind turbulent power spectra at high frequency repeatedly obey spectral indices very close to this number. Boldly referring to Taylor's hypothesis where $\omega_s \propto k$, one might conclude then that a convective flow maps this spectral range of the *advected* turbulent K-spectrum into the spacecraft frame where it appears as an $\omega_s^{-\frac{8}{3}}$ spectrum.

---

[5]The notion of a "turbulent dispersion relation" is alien to turbulence theory which refers to stationary turbulence, conveniently collecting any temporal changes under the loosely defined term intermittency. However, observation of stationary turbulence shows that eddies come and go on an internal timescale, which stationary theory integrates out. In Fourier representation this corresponds to an integration of the spectral density $S(\omega_k, \boldsymbol{k})$ with respect to frequency $\omega_k$ (cf., e.g., Biskamp, 2003) which leaves only the wavenumber dependence. The spectral density $S$ occupies a volume in $(\omega, \boldsymbol{k})$-space. Resolved for $\omega = \omega_k(\boldsymbol{k})$ it yields a complex multiply connected surface, the "turbulent dispersion relation", which has nothing in common with a linear dispersion relation resulting

from the solution of a linear eigenmode wave equation. It contains the dependence of Fourier frequency $\omega_k$ on Fourier wavenumber $\boldsymbol{k}$. Though this should be common sense, we feel obliged to note this here because of the confusion caused when speaking about a "dispersion relation" in turbulence.

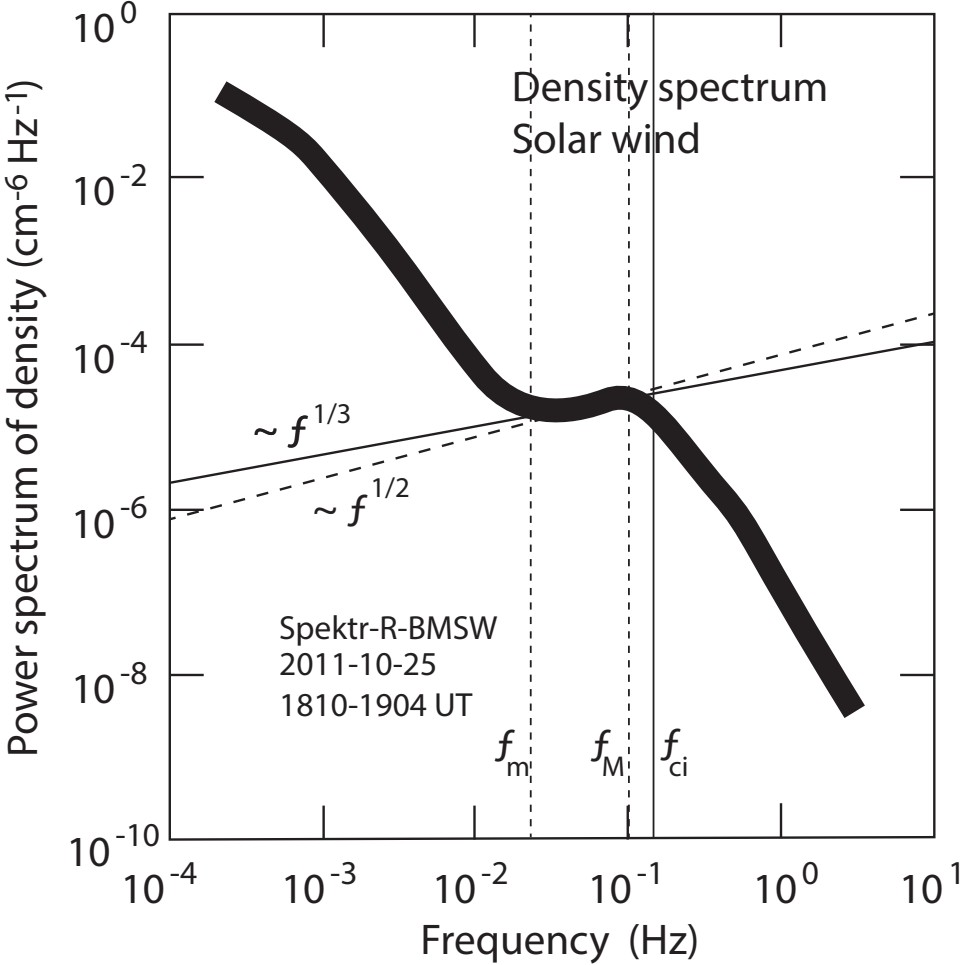

**Figure 3.** Solar wind power spectra of turbulent density fluctuations (based on BMSW data from Šafránková et al., 2013, obtained on Oct 25, 2011). Single point measurements were obtained with six Faraday cups with time resolution of 31 ms ($\sim 30$ Hz) under the following solar wind conditions: density $N \sim 3 \times 10^6$ m$^{-3}$, mean magnetic field $B_0 \sim 8$ nT, bulk speed $V_0 \sim 540$ km/s, ion temperature $T_i \sim 10$ eV, Alfvén Mach number $M_A \sim 6$, total $\beta \sim 0.3$, implying dilute low $\beta$ – high $M_A$ – moderately fast flow conditions. The local thermal ion gyroradius is $\rho_i \sim 2.2 \times 10^4$ m. The vertical indicates the local ion cyclotron $f_{ci} = \omega_{ci}/2\pi \approx 0.15$ Hz frequency. Plasma frequency is $f_i = \omega_i/2\pi \approx 400$ Hz. $f_m, f_M$ are the approximate minimum and maximum frequencies of the bumpy range. The data were averaged over $\sim 1200$ s measuring time and subsequently filtered (cf. Šafránková et al., 2016, for the description of the data reduction). The spectrum shown is the average spectrum with line width roughly corresponding to the largest spread of the filtered data in the logarithmic ordinate direction and applied to the whole spectrum. The power spectrum exhibits a (so-called) bump at intermediate frequencies of positive slopes $\sim \omega^{\frac{1}{3}}$ or $\sim \omega^{\frac{1}{2}}$. This is in agreement with it being caused by the response of the nonmagnetic ions to the electric induction field of the turbulent mechanical fluctuations in the solar wind velocity in Kolmogorov (K, solid line) or Iroshnikov-Kraichnan (IK, dashed line) inertial range turbulence. The large scatter in the data (weight of line) inhibits distinguishing between K and IK inertial range velocity turbulence.

If this is true, then the corresponding observed spectral transition (or break point) from the spectral K-index $\sim \frac{5}{3}$ to the steeper index $\sim \frac{8}{3}$ observed in the large-wavenumber power spectra indicates the division between large-scale energy-carrying, energy-rich turbulent eddies and the bulk of energy-poor small-scale eddies in the mechanical turbulence. It thus provides a simple explanation of the change in spectral index from $\sim \frac{5}{3}$ (K spectrum) to $\lesssim 3$ (advected K turbulence spectrum) without invoking any sophisticated turbulence theory or else as well as no effects of dissipation.

Inspecting the behaviour in the long wavelength range, one finds that the exponential dependence $\exp(-\ell_K^2/U_0^2 k^{\frac{2}{3}})$ suppresses the spectrum here. This flattens the inertial range spectrum towards small wavenumbers $k_{in}$ into the large eddy range where it causes bending of the spectrum. The wavenumber at spectral maximum is

$$k_{min} \lesssim \ell_K^3/16U_0^3\sqrt{2} \tag{17}$$

Approaching from Kolmogorov inertial range towards smaller $k$, one observes flattening until $k_{min} < k_{in}$. In most cases this point will lie outside the observation range.

In the stationary turbulence frame the frequency spectrum is obtained when integrated with respect to $k$ (Biskamp, 2003). It then maps the Doppler broadened advected velocity power spectrum (Fung et al., 1992; Kaneda, 1993) to the Kolmogorov law in the source region frequency space:

$$\int\limits_{k_{in}}^{k_d} dk\, \mathcal{E}^{ad}_{k\omega_k} \sim \mathcal{E}^{ad}_K(\omega) \propto \omega^{-\frac{5}{3}} \tag{18}$$

This mapping is independent on Taylor's hypothesis. It applies strictly only to the turbulent reference frame. When attempting to map it into the spacecraft frame via Taylor's-Galilei transformation, referring to solar wind flow at finite $V_0 \neq 0$, one must return to its wave number representation Eq. (14). This transformation, though straightforward, is obscured by the appearance of $k$ in the exponential through $\omega_\pm$. According to Taylor the turbulence frame frequency transforms like

$$\omega_k = \omega_s - kV_0\cos\alpha. \qquad \alpha = \angle(\boldsymbol{k}, \boldsymbol{V}_0) \tag{19}$$

This is Taylor's Galilei transformation. Neglecting $\omega_k$ implies $\omega_s = kV_0\cos\alpha$. The exponential reduces to

$$\exp\left[-\frac{1}{4}\left(\frac{\omega_k + kV_0\cos\alpha \pm \ell_K k^{\frac{2}{3}}}{kU_0}\right)^2\right] = \tag{20}$$

$$= \exp\left[-\frac{1}{4(k\lambda_i)^{\frac{2}{3}}}\left(\frac{\lambda_i^{\frac{1}{3}}\ell_K}{U_0}\right)^2\right] \tag{21}$$

$$\longrightarrow 1 - \frac{1}{4(k\lambda_i)^{\frac{2}{3}}}\left(\frac{\lambda_i^{\frac{1}{3}}\ell_K}{U_0}\right)^2 \tag{22}$$

$\lambda_i^{\frac{1}{3}}\ell/U_0 \equiv U_\ell^K/U_0$ is a velocity ratio. The arrow holds for the ion-inertial range $k\lambda_i > 1$ and $U_\ell^K/U_0 < 1$. The exponential expression leads to an advected K spectrum as observed by the spacecraft in frequency space

$$\mathcal{E}^{ad}_{\omega_s} \propto \omega_s^{-\frac{8}{3}}\exp\left[-\frac{1}{4}\left(\frac{V_0\cos\alpha}{\omega_s\lambda_i}\right)^{\frac{2}{3}}\left(\frac{U_\ell^K}{U_0}\right)^2\right] \tag{23}$$

which, as before for large $\omega_s$, is of spectral index $\frac{8}{3}$. With decreasing spacecraft frequency $\omega_s$, the exponential correction factor acts suppressing on the spectrum. This corresponds to a spectral flattening towards smaller $\omega_s$. It might even cause a spectral dip, depending on the parameters and velocities involved. The effect is strongest for aligned streaming and eddy wavenumber. For $\alpha \sim 90°$ one recovers the index $\frac{8}{3}$.

It is most interesting that spectral broadening, when transformed into the spacecraft frame in streaming turbulence, causes a difference that strong between the original Kolmogorov and the advected Kolmogorov spectrum. This spectral behaviour is still independent on the Poisson modification which we are going to investigate in the next section.

## 3 Ion-inertial-range density power spectrum

Here we apply the Poisson modified expressions to the theoretical inertial range K and IK turbulence models. We concentrate on the inertial range K spectrum and rewrite the result subsequently to the IK spectrum.

### 3.1 Inertial range K and IK density power spectrum

For the simple inertial range K spectrum we have from Eq. (7) and Eq. (13) that

$$\langle|\delta N|^2\rangle_{k_\perp} = C_K\left(\frac{\epsilon_0 B_0}{e}\right)^2 \epsilon^{\frac{2}{3}} k_\perp^{\frac{1}{3}} \qquad \text{for} \qquad k_{\perp in} < k_\perp < k_{\perp d} \tag{24}$$

This is a very simple wavenumber dependence of the power spectrum of density turbulence, permitting (Treumann et al., 2019) Taylor-Galilei transformation into the spacecraft frame. Setting $k_\perp = \omega_s/V_0\cos\alpha$ we immediately obtain that

$$\langle|\delta N|^2\rangle_{k_\perp} \propto \omega_s^{\frac{1}{3}} \tag{25}$$

with factor of proportionality $C_K(\epsilon_0 B_0/e)^2(\epsilon^2/V_0\cos\alpha)^{\frac{1}{3}}$.

Following exactly the same reasoning when dealing with the IK spectrum, which has power index $\frac{3}{2}$, we obtain that

$$\langle|\delta N|^2\rangle_{k_\perp} \propto \omega_s^{\frac{1}{2}} \tag{26}$$

Hence, the effect of the Poisson response of the plasma to the inertial range power spectra of K and IK turbulence in the velocity is to generate a positive slope in the density power spectrum when Taylor-Galilei transformed into the spacecraft frame.

We now proceed to the investigation of the effect of advection.

### 3.2 Advected Poisson modified spectrum at $V_0 = 0$

Use of the advected power spectral density Eq. (14) of the velocity field for $V_0 = 0$ in the transformed Poisson equation, with $k \to k_\perp$ perpendicular to the mean magnetic field $\boldsymbol{B}_0$, yields for the non-convected advected turbulent ion-inertial range Poisson-modified density-power spectrum in the stationary large-eddy turbulence frame,

$$\begin{aligned} \langle|\delta N|^2\rangle^{ad}_{\omega_k k_\perp} &= \frac{\epsilon_0^2 B_0^2}{e^2} k_\perp^2 \langle|\delta\boldsymbol{V}|^2\rangle_{\omega_k k_\perp} \\ &= \frac{\epsilon_0^2 B_0^2}{e^2} k_\perp^2 \mathcal{E}^{ad}_{k_\perp\omega_k} \\ &\propto k_\perp^{-\frac{2}{3}}\sum_\pm \exp\left[-\frac{1}{2}\frac{\omega_\pm^2}{(kU_0)^2}\right] \end{aligned} \tag{27}$$

Integration with respect to $k_\perp$ under the above assumption on $\omega_\pm \approx k_\perp U_0$ yields for the Eulerian (Fung et al., 1992)

density power spectrum in frequency space $\omega_\ell < \omega < \omega_u$ in the ion-inertial domain of the turbulent inertial range :

$$\langle |\delta N|^2 \rangle_\omega^{ad} \sim \omega^{\frac{1}{3}}, \qquad k_{ir}^{\frac{2}{3}} \epsilon^{\frac{1}{3}} = \omega_\ell < \omega < \omega_u \qquad (28)$$

This is the proper frequency dependence of the advected turbulent density spectrum *in the turbulence frame*. Here $k_{ir} \approx 2\pi\omega_i/c$ (or $2\pi v_i/\omega_{ci}$) is the wavenumber presumably corresponding to the lower end of the ion inertial range. The upper bound on the frequency $\omega_u$ remains undetermined. One assumption would be that $\omega_u$ is the lower hybrid frequency which is intermediate to the ion and electron cyclotron frequencies. At this frequency electrons become capable of discharging the electric induction field thus breaking the spectrum to return to its Kolmogorov slope at increasing frequency.

In contrast to the Kolmogorov law the *Poisson-mediated proper advected* density power spectrum Eq. (31) increases with frequency in the proper stationary frame of the turbulence. This increase is restricted to that part of the inertial K range which corresponds to the ion-inertial scale and frequency range.

The case of an IK spectrum leads to an advected velocity spectrum

$$\langle |\delta \boldsymbol{V}|^2 \rangle_{\omega_k k_\perp} \propto k_\perp^{-\frac{3}{2}} \qquad (29)$$

which yields

$$\langle |\delta N|^2 \rangle_{\omega_k k_\perp}^{ad} \propto k_\perp^{-\frac{1}{2}} \sum_{\pm} \exp\left[ -\frac{1}{2} \frac{\omega_\pm^2}{(kU_0)^2} \right], \qquad (30)$$

$$\omega_\pm = \omega_k \pm \ell_{IK} k_\perp^{\frac{3}{4}}$$

Integration with respect to $k_\perp$ then gives the proper advected frequency spectrum in the stationary frame of IK turbulence

$$\langle |\delta N|^2 \rangle_\omega^{ad} \sim \omega^{\frac{1}{2}} \qquad (31)$$

This proper IK density spectrum increases with frequency like the root of the proper frequency.

### 3.3 Taylor-Galilei transformed Poisson modified advected spectra

Turning to the fast streaming solar wind we find with $k_\perp = \omega_s/V_0\cos\alpha$ for the Poisson modified advected and convected K density spectrum

$$\langle |\delta N|^2 \rangle_{\omega_s}^{K,ad} \propto \omega_s^{-\frac{2}{3}} \exp\left[ -\frac{1}{4}\left( \frac{V_0\cos\alpha}{\omega_s\lambda_i} \right)^{\frac{2}{3}} \left( \frac{U_\ell^K}{U_0} \right)^2 \right] \qquad (32)$$

where we again neglected the proper frequency dependence. This Taylor-Galilei transformed density spectrum decays with increasing frequency albeit at a weak power $\sim \frac{2}{3}$. At large frequency $\omega_s$ the exponential is one, and the spectrum

becomes $\propto \omega_s^{-\frac{2}{3}}$. Towards smaller $\omega_s$ the spectrum flattens and assumes its maximum at

$$\omega_{sm}^K = \frac{3}{8}\left( \frac{V_0\cos\alpha}{\lambda_i} \right)\left( \frac{U_\ell^K}{U_0} \right)^3 \qquad (33)$$

The same reasoning produces for the Poisson modified advected IK spectrum the Taylor-Galilei transformed spacecraft frequency spectrum

$$\langle |\delta N|^2 \rangle_{\omega_s}^{IK,ad} \propto \omega_s^{-\frac{1}{2}} \exp\left[ -\frac{1}{4}\left( \frac{V_0\cos\alpha}{\omega_s\lambda_i} \right)^{\frac{3}{4}} \left( \frac{U_\ell^{IK}}{U_0} \right)^2 \right] \qquad (34)$$

Both advected K and IK spectra have negative slopes in spacecraft frequency $\omega_s$. Like in the case of a K spectrum, this spectrum approaches its steepest slope $\frac{1}{2}$ at large spacecraft frequencies $\omega_s$, while in the direction of small frequencies it flattens out to assume its maximum value at

$$\omega_m^{IK} = \left( \frac{7}{8}\frac{V_0\cos\alpha}{\lambda_i} \right)^{\frac{3}{2}} \left( \frac{U_\ell^{IK}}{U_0} \right)^3 \qquad (35)$$

In both cases of advected K and IK spectra the Taylor-Galilei transformation from the proper frame of turbulence into the spacecraft frame is permitted because it applies to the velocity and density spectra (Treumann et al., 2019). It maps the wavenumber spectrum into the spacecraft frame frequency spectrum. However, in both cases we recover frequency spectra which decrease with frequency though weakly approaching steepest slope at large frequencies. They flatten out towards low frequencies and may assume maxima if only these maxima are still in the inertial range of the advected K or IK spectrum. Only in this case the spacecraft frequency spectrum exhibits a bump at their nominal maximum frequencies $\omega_{sm}$. Whence the maximum frequency falls outside the ion-inertial range the bump will be absent, while the spectrum will be flatter than at large frequencies. Such flattened bumpless spectra have been observed. The next subsections provides examples of observed bumpy and bumpless spectra in the spacecraft frequency frame.

## 4 Application to selected observations in the solar wind

In the following two subsections we apply the above theory to real observations made in situ in the solar wind. We first consider density power spectra exhibiting well expressed spectral bumps of positive slope. We then show two examples where no bump is present but the power spectra exhibit a scale limited excess and consequently a scale limited spectral flattening.

### 4.1 Observed bumpy solar wind power spectra of turbulent density

Figure 3 is an example of a density spectrum with respect to spacecraft frequency which exhibits a positive slope (or

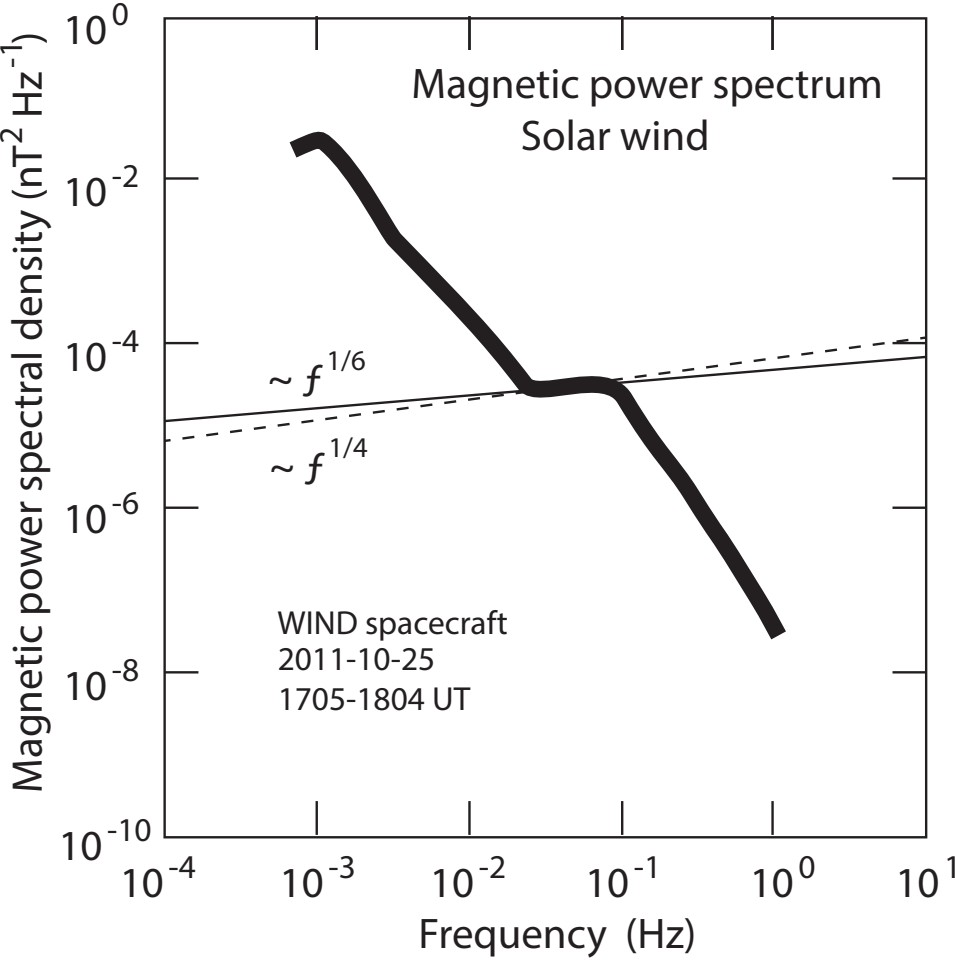

**Figure 4.** Solar wind power spectra of the turbulent magnetic field for the same time interval as in Figure 3 measured by the WIND spacecraft (data from Šafránková et al., 2013) which was located in the Lagrange point L1. Line width accounts for the scatter of data. The magnetic turbulence spectrum exhibits a deformation similar to that in the density power spectrum and same frequency interval. The positive slope $\sim \omega^{\frac{1}{6}}$ in the deformation confirms its origin from pressure balance. It indicates its nature being secondary to turbulence in density. The solid (dashed) line corresponds to an K (IK) velocity spectrum. The scatter of data was again substantial, thus inhibiting distinction between the two cases.

bump) on the otherwise negative slope of the main spectrum. The data in this figure were taken from published spectra (Šafránková et al., 2013) in the solar wind at an average bulk velocity fo $V_0 \approx 534$ km s$^{-1}$, density $N_0 \approx 3 \times 10^6$ m$^{-3}$, magnetic field $B_0 \approx 8$ nT, yielding a super-alfvénic Alfvén Mach number $M_A \approx 6$, ion temperature $T_i \lesssim 3$ eV, and total plasma $\beta \approx 0.3$, i.e. low-beta conditions. The straight solid and broken lines drawn across this slope correspond to the predicted $\sim \omega^{\frac{1}{3}}$ K and $\sim \omega^{\frac{1}{2}}$ IK slopes under convection dominated conditions. Both these lines fit the shape very well though it cannot be decided which of the inertial range turbulence models provides a better fit, as the large scatter of the data mimicked by the line width inhibits any distinction. It is however obvious from Table 1 that advection plays no role in this case.

In order to check pressure balance between the density and magnetic field fluctuations, we refer to turbulent magnetic power spectra obtained at the WIND spacecraft Šafránková et al. (2013). WIND was located in the L1 Lagrange point. Magnetic field fluctuations were related in time to the BMSW observations by the solar wind flow. In spite of their scatter, the data were sufficiently stationary for comparison to the density measurements.

Figure 4 shows the Wind magnetic power spectral densities. For transformation of the point cloud into a continuous line we applied the same technique (Šafránková et al., 2016) as to the density spectrum. The spectrum exhibits the expected positive slope in the BMSW frequency interval. The straight solid and broken lines along the positive slope correspond (within the uncertainty of the observations) to

**Table 1.** K and IK ion inertial range spectral indices $k^{-a}$, $k^{-(a-2)}$, $\omega_s^b$, $\mathcal{E}_{Bs} \sim \omega_s^{b/2}$ without and with advection

| $\langle|\delta V|^2\rangle$ | $a$ | $a-2$ | $b$ | $b/2$ |
|---|---|---|---|---|
| $\mathcal{E}_K$ | $\frac{5}{3}$ | $-\frac{1}{3}$ | $\frac{1}{3}$ | $\frac{1}{6}$ |
| $\mathcal{E}_K^{ad}$ | $\frac{8}{3}$ | $\frac{2}{3}$ | $-\frac{2}{3}$ | $-\frac{1}{3}$ |
| $\mathcal{E}_{IK}$ | $\frac{3}{2}$ | $-\frac{1}{2}$ | $\frac{1}{2}$ | $\frac{1}{4}$ |
| $\mathcal{E}_{IK}^{ad}$ | $\frac{5}{2}$ | $\frac{1}{2}$ | $-\frac{1}{2}$ | $-\frac{1}{4}$ |

the root-slopes of K and IK density inertial range spectra $\langle|\delta B|^2\rangle_{\omega_s} \sim \omega_s^{\frac{1}{6}}$ respectively $\sim \omega_s^{\frac{1}{4}}$. The magnetic spectrum is the consequence of the K or IK density spectrum $\sim \omega_s^{\frac{1}{3}}$ respectively $\sim \omega_s^{\frac{1}{2}}$. Fluctuations in temperature do, within experimental uncertainty, not play any susceptible role. Comparing absolute powers is inhibited by the ungauged differences in instrumentation. (One may note that power spectral densities are positive definite quantities. Measuring their slopes is sufficient indication of pressure balance. Detailed pressure balance can only be seen when checking the phases of the fluctuations. Density and magnetic field would then be found in antiphase.)

## 4.2 The normal case: flattened density power spectra without bump

The majority of observed density power spectra in the solar wind do not exhibit positive slopes. Such spectra are of monotonic negative slope. In this sense they are normal. They frequently possess break points in an intermediate range where the slopes flatten. Two typical examples are shown in Fig. 5 combined from unrelated BMSW and WIND data (Šafránková et al., 2013; Podesta & Borovsky, 2010).

Their flattened spectral intervals each extend roughly over one decade in frequency. The BMSW spectrum is shifted by one order of magnitude in frequency to higher frequencies than the WIND spectrum. Its low-frequency part below the ion-cyclotron frequency $f < f_{ci}$ has slope $\sim \omega^{-\frac{7}{4}}$ close to a K-spectrum $\sim \omega^{-\frac{5}{3}}$. The slope of the flat section is $\sim \omega^{-1}$ which is about the same as the slope of the entire low frequency WIND spectrum before its spectral break. None of the Poisson-modified K or IK spectral slopes fit this flattened regions. At higher frequencies the BMSW spectrum steepens and presumably enters the dissipative range.

The slope of the WIND spectrum above its break point at frequency $\sim 10^{-2}$ Hz decreases to $\sim \omega^{-\frac{1}{2}}$. This corresponds perfectly to an advected Taylor-Galilei transformed IK spectrum, suggesting that WIND detected such a spectrum in the ion-inertial range which maps to those spacecraft frequencies. The pronounced $\omega^{-1}$ spectrum at lower frequencies remains, however, unexplained for both spacecraft.

When crossing the cyclotron frequency $f_{ci}$ the WIND spectrum steepens. We also note that the normalised power spectral densities of WIND at $\langle|\delta N|^2\rangle/N_0^2 > 0.3$ and BMSW at $0.005 < \langle|\delta N|^2\rangle/N_0^2 < 0.05$ in the common slope $\sim \omega^{-1}$ interval are roughly two orders of magnitude apart. This can hardly be traced back to the radial difference of 0.01 AU between L1 and 1 AU.

The obvious difference between the two plasma states is not in the Mach numbers but rather in $\beta$ and $V_0$. BMSW observed under moderately high-$\beta$-low $V_0$, WIND under moderately low-$\beta$-high $V_0$ conditions at similar densities and Mach numbers. Because of the Galileian relation $k = \omega_s/V_0\cos\alpha$, the high speed in the case of WIND seems responsible for the spectral shift of the $\omega_s^{-1}$ spectral range to lower than BMSW frequencies. This, however comes up merely for a factor 2 which does not cover the frequency shift of more than one order of magnitude. Rather it is the angle between mean speed and wavenumber spectrum which displaces the spectra in frequency. If this is the case, then the WIND spectrum was about parallel to the solar wind velocity with WIND angle $\alpha \approx 0°$, while the BMSW spectrum was close to perpendicular with angle $\alpha \approx 90°$, and it is the BMSW spectrum which has been Taylor-Galilei shifted into the high-frequency domain, while the WIND spectrum is about original. This may also be the reason why BMSW does not see the narrow flattened spectral part while compressing the $\omega_s^{-1}$ part into just one order of magnitude in frequency. The near perpendicular angle $\alpha$ will be confirmed below also in the bumpy BMSW spectral case.

## 5 Discussion

In this communication we dealt with the power spectra of density in low frequency plasma turbulence. We did not develop any new theory of turbulence. We showed that, in the ion-inertial scale range of non-magnetised ions, the electric response of the ion population to a given theoretical turbulent K or IK spectrum of velocity may contribute to a scale-limited excess in the density fluctuation spectrum with positive or flattened slope. We demonstrated that the obtained inertial range spectral slopes within experimental uncertainty are not in disagreement with observations in the solar wind, but we could not decide between the models of turbulence. This may be considered a minor contribution only, it shows however, that correct inclusion of the electrodynamic transformation property is important and suffices to reproduce an observational fact without any need to invoke higher order interactions, nor any instability, nor nonlinear theory. We also inferred the limitations and scale ranges for the response to cause an effect. However, a substantial number of unsolved problems remain. Below we discuss some of them.

### 5.1 Reconciling the spectral range

The main problem concerns the agreement with observations. Determination and confirmation of spectral slopes is

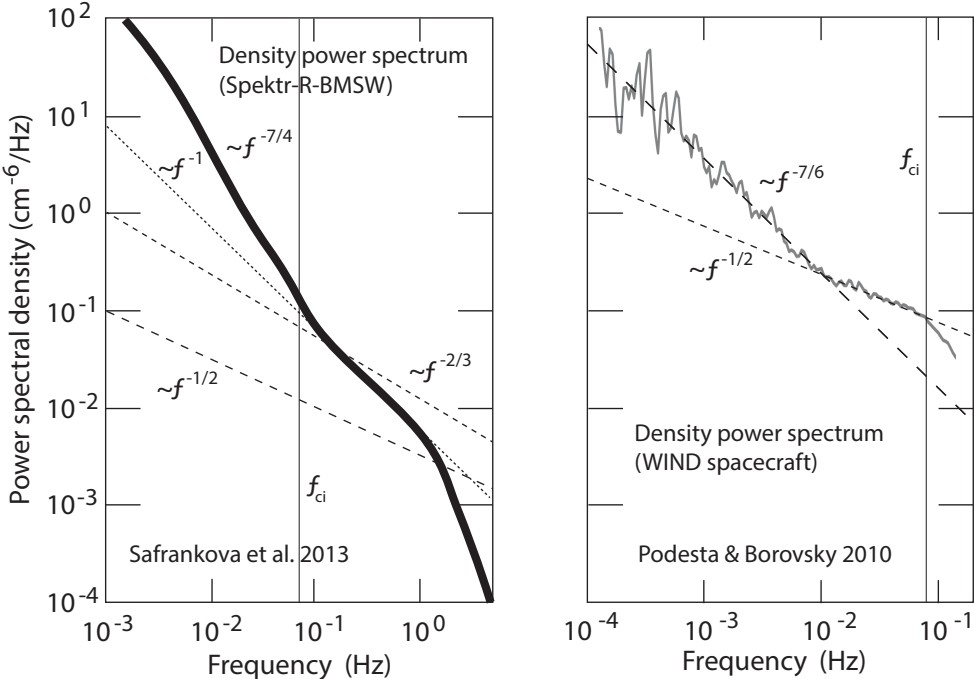

**Figure 5.** Two (redrawn on same scale) cases of "normal" solar wind density power spectra measured by Spektr-R-BMSW (Šafránková et al., 2013) on Nov 10, 2011 and WIND (Podesta & Borovsky, 2010) on Jan 4-8, 1995 at different solar wind conditions. BMSW observations of 2011 were obtained under low speed ($\sim 370$ km/s) moderately large total $\beta = \beta_i + \beta_e \sim 2.5$, high Alfvénic Mach number $M_A \sim 10$, main field $B_0 \sim 5$ nT conditions. Density and temperature amounted to $N_0 \sim 5 \times 10^6$ m$^{-3}$ and $T_i \sim 10$ eV, with ion cyclotron $f_{ci} \sim 0.08$ Hz and plasma $f_i \sim 500$ Hz frequencies. WIND observations in L1 were obtained under high speed ($\sim 640$ km/s), $\beta_i \lesssim 1$, $B_0 \sim 6$ nT, $N_0 \sim 3.5 \times 10^6$ m$^{-3}$, $T_i \sim 20$ eV, $M_A \sim 9$ conditions with similar cyclotron and plasma frequencies. In contrast to Figure 3 these spectra do not exhibit regions of positive slope. Their spectral slope is interrupted by a flattened region. They share a range of spectral index $\sim -1$, though in different frequency intervals, while the WIND spectrum exhibits a higher-frequency range of flat slope $\sim -\frac{1}{2}$ which is absent in the BMSW spectrum.

a necessary condition. However, how to adjust the observed frequency range?

Inspecting Fig. 3 where we included the local ion cyclotron frequency $f_{ci} = \omega_{ci}/2\pi$ finds the scale-limited posi-
tive slope (bump) of the density power spectrum at spacecraft frequencies $f_m \sim \omega_m < \omega_{ci} \sim 0.22$ Hz. According to Taylor we have

$$\omega_m = k_m V_0 \cos \alpha_m, \quad \text{and also} \quad k_m \lambda_i > 1 \qquad (36)$$

where $\alpha_m$ is the angle between $\boldsymbol{k}_m$ and velocity $\boldsymbol{V}_0$, and
$\lambda_i = c/\omega_i$. The first expressions yields a Taylor-Galilei transformed wavenumber $k_m \sim 2.3 \times 10^{-6}/\cos \alpha_m$ m$^{-1}$ From the second we have with the observed ion plasma frequency $k_m \sim 2\pi \times 10^{-6}$ m$^{-1}$. Hence we find that $\cos \alpha_m < 0.37$ or $\alpha_m > 69°$. The turbulent eddies are at highly oblique angles
with respect to the flow velocity.

With angles of this kind the positive slope spectral range can be explained. The lower frequencies then correspond to eddies which propagate nearly perpendicular. Since our theory generally restrict to wave numbers perpendicular to
the ambient magnetic field, the eddies which contribute to

the bumps are perpendicular to $\boldsymbol{B}_0$ and highly oblique with respect to the flow. Similar arguments apply to the high-frequency excess in the WIND observations of Fig. 5. Referring to Table 1 this excess is explained as survival of the advected spectrum when Taylor-Galilei transformed into the spacecraft frame.

## 5.2 Radially convected spectra: Effect of inhomogeneity

The assumption of Taylor-Galilei transformation in the way we used it (and is generally applied to turbulent solar wind power spectra) is valid only in stationary homogeneous turbulent flows of spatially constant plasma and field parameters,[6] which in the solar wind is not the case. It also assumes that wave numbers $k$ are conserved by the flow.[7] Thus the above conclusion is correct only, if the turbulence is gen-

---

[6]For general restrictions on its applicability already in homogeneous MHD see Treumann et al. (2019).

[7]This is a strong assumption. In the absence of dissipation, individual frequencies are conserved. They correspond to energy. Wave

erated locally and is transported over a distance where the radial variation of the solar wind is negligible. If it is assumed that the turbulence is generated in the innermost heliosphere at a fraction of 1 AU (cf., e.g., McKenzie et al., 1995), any simple application of Taylor's-Galilei transformation and thus the above interpretation break down.

Under the fast flow conditions of Fig. 3 it is reasonable to assume that the solar wind expands isentropically, denoting the turbulent source and spacecraft locations by indices $q, s$, respectively. The turbulent inertial range is assumed collisionless, dissipationless, and in ideal gas conditions. For simplicity assume that the expansion is stationary and purely radial. Under Taylor's assumption each eddy maintains its identity, which implies that the number of eddies is constant, and the eddy flux $F_s(r_s)/F_q(r_q) = r_q^2/r_s^2$, i.e. the turbulent power decreases as the square of the radius. For the plasma we have the isentropic condition (e.g., Kittel & Kroemer, 1980, p. 174)

$$\frac{T_s(r_s)}{T_q(r_q)} = \left[\frac{N_s(r_s)}{N_q(r_q)}\right]^{\gamma-1}, \qquad \gamma = \tfrac{5}{3} \tag{37}$$

which gives $N_s(r_s)/N_q(r_q) = \left(r_q/r_s\right)^3$, and thus $T_s(r_s)/T_q(r_q) = \left(r_q/r_s\right)^2$. One requires that $k_q > \lambda_{iq}^{-1} = \omega_{iq}(r_q)/c$. By the same reasoning as in the homogeneous case one finds that

$$\begin{aligned}\frac{f_m}{f_{is}} &= \frac{k_m V_0}{\omega_{is}} \cos\alpha_m \\ &= k_q \lambda_{iq} \frac{V_0}{c}\left(\frac{r_q}{r_s}\right)^{\frac{3}{2}} \cos\alpha_m \\ &\gtrsim \frac{V_0}{c}\left(\frac{r_q}{r_s}\right)^{\frac{3}{2}} \cos\alpha_m\end{aligned} \tag{38}$$

inserting for the left hand side and $V_0$ we find with $r_s = 1$ AU that

$$r_q < \frac{0.1}{\left(\cos\alpha_m\right)^{\frac{2}{3}}}\ \text{AU} \tag{39}$$

We conclude that under the assumption of isentropic expansion of the solar wind and Taylor-Galilei transport of turbulent eddies from the source region to the observation site at 1 AU the generation region of the turbulent eddies which contribute to the bump in the K or IK density power spectrum must be located close to the sun. The marginally permitted angle $\alpha_m$ between wave number and mean flow is obtained putting $r_q = 1$ AU, yielding $\alpha_m > 47°$, meaning that the flow must be oblique for the effect to develop, a conclusion already found above for homogeneous flow. These numbers are obtained under the unproven assumption that Taylor-Galilei transport conserves turbulent wave numbers in the inhomogeneous solar wind.

---

numbers correspond to momenta which do not obey a separate conservation law.

### 5.3 Ion gyroradius effect

So far we referred to the inertial length as limiting the frequency range. We now ask for the more stringent condition $k\rho_{ic} > 1$ that the responsible length is the ion gyroradius $\rho_{ic} = v_i/\omega_{ci}$. In this case reference to the adiabatic conditions becomes necessary. We also need a model of the radial variation of the solar wind magnetic field. The field inside $r_s = 1\,\text{AU}$ is about radial. Magnetic flux conservation yields the Parker model $B_s(r_s) = B_q(r_q)(r_q/r_s)^2$. A more modern empirical model instead proposes a weaker radial decay of power $\frac{5}{3}$ (for a review cf., e.g., Khabarova, 2013). With these dependences we have for

$$\begin{aligned}k\rho_s(r_s) &= k\rho_q(r_q)\frac{B_q(r_q)}{B_s(r_s)}\sqrt{\frac{T_{is}(r_s)}{T_{iq}(r_q)}} \tag{40}\\ &= k\rho_q(r_q)\left(\frac{r_s}{r_q}\right)^{\frac{2}{3}} > \left(\frac{r_s}{r_q}\right)^{\frac{2}{3}} \tag{41}\end{aligned}$$

where the necessary condition $k\rho_q > 1$ has been used. Referring again to the observed minimum frequency $f_m$ yields

$$\begin{aligned}\frac{f_m}{f_{ic,s}} &= \frac{k_m V_0}{\omega_{ic,s}}\cos\alpha_m \tag{42}\\ &= k\rho_q \frac{V_0}{v_i}\left(\frac{r_s}{r_q}\right)^{\frac{2}{3}}\cos\alpha_m \gtrsim \frac{V_0}{v_i}\left(\frac{r_s}{r_q}\right)^{\frac{2}{3}}\cos\alpha_m \tag{43}\end{aligned}$$

Inserting for the frequency ratio $f_m/f_{ic,s} \sim 0.1$ and the ratio of mean to thermal velocities $V_0/v_i \approx 18$, and setting $r_s = 1$ AU we obtain for the source radius lying inside 1 AU

$$1\ \text{AU} > r_{qm} > 300\left(\cos\alpha_m\right)^{\frac{3}{2}} \tag{44}$$

which gives the result $\alpha_m \gtrsim 89°$ for the propagation angle obtained above. According to both these estimates eddy propagation is required to be quasi-perpendicular to the flow. This holds under the strong condition that the wave number is conserved during outward propagation.

### 5.4 Radial variation of wavenumber in expanding solar wind

The wave number $k \sim \lambda^{-1}$ is an inverse wavelength. Let us assume that $\lambda \sim r$ stretches linearly when the volume expands, thereby reducing $k$ hyperbolically. The eddies, which are frozen to the volume, also stretch linearly. In this case the ratio $r_s/r_q$ in Eq. (43) is raised to the power $\frac{1}{3}$, and we find instead that

$$r_{qm} \lesssim \left(\frac{\cos\alpha_m}{18}\right)^3 < 1\ \text{AU} \tag{45}$$

This gives $\alpha_m \gtrsim 87°$ which is not too different from the above case. Thus the angle between mean speed and turbulent wavenumber is close to perpendicular in order to reconcile the lower observed limit in spacecraft frequency with the wavenumber in the source region.

## 5.5 High-frequency limit for $f_M \sim f_{ce,s}$

A similar reasoning can be applied to the upper frequency bound $\omega_M$. Following the discussion in the Introduction, this bound is caused by the truncation of the ion inertial range at large wavenumbers when the scale approaches the electron scale, electron inertia takes over, and electrons demagnetise. The condition in this case is that $k\rho_e < 1$, which defines the maximum frequency $\omega_M$.

We then have the following relation for the maximum wave number:

$$k\rho_{Me}(r_s) = k\rho_{Me}(r_q)\frac{B_q(r_q)}{B_s(r_s)}\sqrt{\frac{T_{es}(r_s)}{T_{eq}(r_s)}} < \left(\frac{r_s}{r_q}\right)^{\frac{2}{3}} \quad (46)$$

From the maximum observed frequency we find with $f_M \lesssim f_{ce,s}$

$$\begin{aligned}\frac{f_M}{f_{ce,s}} &= \frac{k_M V_0}{\omega_{ce,s}}\cos\alpha_M \\ &= k\rho_q \frac{V_0}{v_e}\left(\frac{r_s}{r_q}\right)^{\frac{2}{3}}\cos\alpha_M \lesssim \frac{V_0}{v_e}\left(\frac{r_s}{r_q}\right)^{\frac{2}{3}} \lesssim 1 \quad (47)\end{aligned}$$

which, when inserting $k\rho_q \lesssim 1$, adopting the main plasma parameters, and with maximum frequency $f_M/f_{ce,s} \sim 1$ and $r_s \sim 1$ AU, yields

$$r_{qM} > \left(\frac{V_0}{v_e}\right)^{\frac{3}{2}} \text{ AU} \sim 0.05 \text{ AU} \quad (48)$$

Taking the two results for this case together the observations map to an angle of propagation $\alpha_m > 49°$ and places the turbulent source close to the sun but outside $11\,\mathrm{R}_\odot \lesssim r_q \lesssim 1$ AU. It occurs only, if the turbulence contains a dominant population of eddies obeying wave number vectors $\boldsymbol{k}$ which are oblique to the mean flow velocity $\boldsymbol{V}_0$. This is in agreement with our above given estimate on the theoretical limits and explains the relative rarity of its observation. Unfortunately, based on the observations, the desired location of the turbulent source region in space cannot be localised more precisely.

## 5.6 The observed case: $f_m \sim 0.1 f_M \ll f_{ce,s}$

Reconciling the observed range of the bump poses a tantalising problem. Our theoretical approach would suggest that the bump develops between the two cyclotron frequencies of ions and electrons in the spacecraft frame. This would correspond to a range of the order of the mass ratio $m_i/m_e$ which would be three orders of magnitude. The actually observed range $f_m \lesssim f_s \lesssim f_M$ is much narrower, just one order of magnitude. Given the uncertainties of measurement and instrumentation this can be extended at most to the root of the mass ratio, which in a proton-electron plasma amounts to a factor of $f_M/f_s \sim 43$ only. In addition, unfortunately, the observed local maximum frequency in Fig. 3 is far less

than the local electron cyclotron frequency $f_M \ll f_{ce,s}$. The affected wavenumber and frequency ranges are very narrow and at the wrong place. Thus in the given version the above reasoning does not apply. Already in the source region the effect must be bound to a narrow domain in wavenumber. The mass ratio might suggest coincidence with the lower-hybrid frequency of a low-$\beta$ proton-electron plasma which, when raised to the power $\frac{3}{2}$, yields

$$r_{qM} \gtrsim 0.6 \text{ AU} \quad (49)$$

putting the source region substantially farther out to $\gtrsim 45$ $\mathrm{R}_\odot$.

The latter estimate is, however, quite speculative. Thus the narrowness of the observed bump in frequency poses a serious problem. Its solution is not obvious. The most honest conclusion is that little can be said about the observed upper frequency termination of the bump in Fig. 3 unless an additional assumption is made.

One may, however, argue that in a high-$\beta_i$ plasma the gyroradius of the ions is large. The ions are non-magnetic, but the effect can arise only when the wavelength becomes less than the inertial length $\lambda_m < \lambda_i = c/\omega_i$. Similarly the effect will disappear whence the wavelength crosses the electron inertial length $\lambda_M < \lambda_e = c/\omega_e$. The ratio of these two limits is $\lambda_M/\lambda_m = f_M/f_m = \sqrt{m_i/m_e} \approx 43$. This agrees approximately with the observation. This interpretation then identifies the range of the effect in spacecraft frequency and source wavenumber with the range between electron and ion inertial lengths. Since both evolve radially with the ratio of the root of densities, the relative spectral width should not change from source to spacecraft.

In order to get an idea of the distance between source and spacecraft, we assume that in the interval between the minimum and maximum frequencies the ion cyclotron frequency is crossed. Hence the corresponding wavenumber is contained in the spectrum though it is invisible. This fact, however, enables to refer to the difference in the ion inertial length scale and the ion gyroradius. The total difference in frequency amounts to roughly one order of magnitude. The ratio of both lengths is $\rho_i/\lambda_i = \sqrt{\beta_i}$ with $\beta_i$ the ion-$\beta$. In isentropic expansion the evolution of $\beta_i$, assuming a Parker model, is

$$\left(\frac{\rho_{is}}{\rho_{iq}}\frac{\lambda_{iq}}{\lambda_{is}}\right)^2 = \frac{\beta_{is}}{\beta_{iq}} \propto \left(\frac{r_q}{r_s}\right)^{\frac{5}{3}} \quad (50)$$

From observations we have a total $\beta > 1$. We expect $\rho_i \gtrsim \lambda_i$, and assume $\beta_i \gtrsim 1$. Figure 3 suggests a frequency ratio $f_m/f_M \sim \beta_{is} \sim 0.7$ larger than $\sqrt{m_e/m_i} \approx 0.025$. The affected frequency and wavenumber ranges are limited from above when the scale approaches the electron gyroradius. In that case the upper bound is not determined by the mass ratio alone. With the measured frequency ratio, the location of the source should then be outside a shortest distance of

$$r_q \gtrsim 0.24\,\beta_{iq} \quad \text{AU} \quad (51)$$

This value corresponds to $> 50\,\beta_{iq}$ R$_\odot$ from the sun. Since the source must lie inside $r_q < 1$ AU, we conclude that $\beta_{iq} \lesssim 4.15$. This number is just an upper limit. It is consistent with model calculations (McKenzie et al., 1995) which predict $\beta_{iq} < 1$ shifting the inner boundary of the turbulent source region further in.

### 5.7  Summary and outlook

In this paper, we considered the cases $V_0 = U_0 = 0$, $V_0 = 0, U_0 \neq 0$, and $V_0 \neq 0$ for K and IK velocity spectra, where $V_0$ is the velocity of the mean solar wind stream, and $U_0$ is the mean speed of the energy-carrying largest turbulent MHD vortices which advect the bulk of small scale turbulence around (Tennekes, 1975). In the K and IK models of turbulence they, in addition, play the role of the energy injectors. The resulting spectral slopes are given as $b$ in Table 1 fourth column. The input spectral power densities are $\mathcal{E}_{IK}, \mathcal{E}_{IK}^{ad}$. Each of them yields a different ion inertial scale range power spectrum in $k$-space and, consequently, also a different power law spectrum in $\omega_s$-space.

Table 1 shows that the ordinary spectra acquire positive slopes in wave number $k$ in the frame of stationary and homogeneous turbulence in the turbulence frame. However, observations of this slope in frequency undermine this conclusion, suggesting that it is the ordinary K (IK) velocity turbulence (or if anisotropy is taken into account, the Kolmogorov-Goldreich-Sridhar KGS) spectrum in the ion-inertial range which, when convected by the solar wind flow across the spacecraft, deforms the density spectrum. All advected spectra have, in contrast, negative slope in frequency which in this form disagrees with observation of the spectral bumps.

The obtained advected slopes in the stationary turbulence frame are also too far away from the flattest notorious and badly understood negative slope $\omega_s^{-1}$ for being related. Their nominal K and IK slopes are $-\frac{2}{3}$ and $-\frac{1}{2}$ respectively. This implies that spacecraft observations interpreted as observing the local stationary turbulence do in their majority not detect an advected convected spectrum in the ion-inertial K (IK) inertial range. They are, however, well capable of explaining the high-frequency flattened spectral excursion in the WIND spectrum which is shown in Fig. 5. It has the correct advective IK spectral index $-\frac{1}{2}$ when convected across the WIND spacecraft before onset of spectral decay.

Generally the form of a distorted power spectrum in density depends on the external solar wind conditions. The reconciliation of these with the theoretical predictions and the observation of the spectral range of the distortion is a difficult mostly observational task. We have attempted it in the discussion section. In particular the proposed bending of the power spectral density in the direction of lower frequencies requires identification of the maximum point of the advected spectrum in frequency and the transition to the undisturbed K or IK inertial ranges.

We also noted that in an expanding solar wind thermodynamic effects must be taken into account, which we tentatively tried in order to obtain some preliminary information about the angle between flow and the turbulent wavenumbers which contribute to deformation of the spectrum. Some tentative information could also be retrieved in this case about the radial solar distance of the turbulent source region. When thermodynamics come in, one may raise the important question for the collisionless turbulent ion heating $\langle \delta \dot{Q}_i \rangle = -\langle \delta \dot{Q}_{em} \rangle = \langle \delta \boldsymbol{J} \cdot \delta \boldsymbol{E} \rangle$ in the ion-inertial range, the negative of the mean loss in electromagnetic energy density per time $\langle \delta \dot{Q}_{em} \rangle$, proportional to the product of current vortices $\delta \boldsymbol{J}$ and the turbulent electric field $\delta \boldsymbol{E}$. Though of finite magnitude, it is second order. This is left for future investigation. Hall currents do not contribute to any heating.

So far we have not taken into account the contribution of Hall spectra. These affect the shape of the density spectrum via the Hall magnetic field, a second order effect indeed though it might contribute to additional spectral deformation. Inclusion of the Hall effect requires a separate investigation with reference to magnetic fluctuations. On those scales the Hall currents should provide a free energy source internal to the turbulence which in K and IK theory is not included.

Hall fields are closely related to kinetic effects in the ion-inertial range. Among them are kinetic Alfvén waves whose perpendicular scales $k_\perp \sim \lambda_i^{-1}$ agree with the scale of the ion-inertial range. Possibly they can grow on the expense of the Hall field which in this case plays the role of free energy for them. If they can grow to sufficiently large amplitudes, they contribute to further deforming I and IK ion-inertial range density spectra.

Similarly, *small-scale shock waves* might evolve at the inferred high Mach numbers when turbulent eddies grow and steepen in the small scale range. These necessarily become sources of electron beams, reflect ions, and transfer their energy in a kinetic-turbulent way to the particle population. Such beams act as sources of particular wave populations which contribute to turbulence preferably at the kinetic scales of interest.

Inclusion of all these effects is a difficult task. It still opens up a wide field for investigation of turbulence on the ion-inertial scale not yet entering the ultimate (Treumann & Baumjohann, 2015) collisionless dissipation scale where electrons de-magnetise as well and the current filaments dissipate their energy in the process of *spontaneous collisionless reconnection* as the most probable ultimate energy sink of otherwise collisionless turbulence. The scales of this dissipation process are still far away from any molecular scales. The resulting dissipation is justifiably anomalous.

*Acknowledgement.* This work was part of a brief Visiting Scientist Programme at the International Space Science Institute Bern. We acknowledge the interest of the ISSI directorate as well as the generous hospitality of the ISSI staff, in particular the assistance of

the librarians Andrea Fischer and Irmela Schweitzer, and the Systems Administrator Saliba F. Saliba. We also thank the anonymous reviewer for intriguing comments and criticism.

*Author contribution.* All authors contributed equally to this paper.
*Data availability.* No data sets were used in this article.
*Competing interests.* The authors declare that they have no conflict of interest.

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
