# Peer review of "On the ion-inertial range density-power spectra in solar wind turbulence"

_Annales Geophysicae, 2018_

## Referee Comment (RC1) · Anonymous Referee #1 · 2 Jan 2019

The manuscript "On the ion-inertial range density power spectra in solar wind turbulence" by Treumann et al. describes a possible explanation for the occasional "bumps" seen in the power spectra density in solar wind measurements around the ion scale. This interpretation is based on the electric ion response due to quasi-neutrality and pressure balance with the magnetic power spectra. The idea is interesting and worth to be published. The derivations seems to be logical from the basic equations and (sometimes) strong assumptions, but my main concern (in General Comment 1) is that the organization of the paper makes it hard to read (for me and maybe other readers), in the sense that is easy to lose focus: many times I did not understand the purpose of some paragraphs or derivations for backing up the main conclusion of this paper. I maybe wrong, but I would appreciate at least the opinion of the authors about this

issue.

The rest of my comments, including some suggestions for improvement, are as follows.

GENERAL COMMENTS:

1) It is not clear to me (and maybe to some future readers of this manuscript) what part of all this discussion of this manuscript (mostly the equations) can be traced back to previous references and what part can be considered original as to the knowledge of the authors. It would be nice to add some references to the equations where appropriate. I understand that they propose a different interpretation for the bump in the density spectrum, but many of the equations seem (to me) to be widely known while others not that much (probably due to some special assumptions). I am not sure where to draw the line in order to say that from this point on it is mostly different from the standard theory of turbulence. It might be due to the organization of this manuscript, it is not that easy to follow, there are some digressions halfway that make difficult to see what is exactly the focus and what is the purpose of each section/paragraph toward the final goal, there is lack of "roadmap". In summary, the presentation and organization of the ideas could be improved.

2) The explanation of the flattening or bump in the density spectra of turbulent fluctuations due to Kinetic Alfvén waves is very popular (see, e.g, [Howes2011] and later reviews by the same author, [Harmon2005], etc). Why is not properly discussed here?. It is fine to put forward an additional explanation, but I do not find fair to neglect a more common one without good arguments or comparison of their respective advantages/disadvantages (KAWs are mentioned only once in this paragraph in page8, "to propagate in the kinetic Alfvén mode, or attributed to dissipation. Though this is neither impossible nor can it be excluded here, the rather more convincing conclusion is that we are dealing with de-magnetised ions in the ion-inertial domain which respond to the turbulent induction electric field and become swept over the spacecraft by the fast solar wind flow" ...and that explanations given there is not actually convincing. And even

more so taking into account the large scatter in the data. )

3) In-situ measurements are not the only way to get density power spectra in the solar wind. Scintillation observations of radio wave propagation can also provide complementary insights and cross-comparisons. That technique is often used to get density spectra in the interstellar medium ISM (mostly through diagnostics of pulsar radio-emission), since scintillation is dominated by small-scale density fluctuations [Armstrong1981]. Interestingly, scintillation observations in the solar wind have also revealed a kind of "bump" (or at least a flattening) in the power density spectra at ion scales [Coles1989, Spangler1995, Harmon2005], while their counterpart in the ISM seems to be missing [Haverkorn2013 and references therein]. It would be interesting to add at least a comment about this in this manuscript.

SPECIFIC COMMENTS:

- page2, lines3-4: "....K or I-K spectra respectively their anisotropic generalisation"—-> This is not clear, is there may be a missing word? (respectively with respect to what?)

- page2, line9: Please define the angular brackets in Eq 1 (those are mentioned much later).

- page2, line30: What is "some unknown strongly damped virtual evanescent mode..."? A wave mode with all those adjectives is very vague, it can mean anything and confuse readers (it confused me, specially with the phrase after that says "...contribute to both density and temperature"). Please be more specific and/or provide references.

- page2, line31: "in the higher frequency range their existence"—-> please specify to what is being referred by "their"

- page3, Fig1, and page6, Figure2. How do exactly the data of those figures was obtained from the data of Safrankova2016? Please provide more details for the sake of clarity.

- page3, line1: "Above frequencies > 10ˆ Hz.....those frequencies exceeding the ion cy-

clotron frequency"—-> Please specify the local ion cyclotron frequency for those in-situ measurements (otherwise it is hard to compare frequencies in Hertz with characteristic plasma frequencies in dimensionless units).

- page4, lines4-5: "...electromotive force terms..... do not vary and, in considering the effects of the electric field on turbulence, can be dropped"—> This does not seem straightforward for me. Could you please be more clear on how that conclusion was reached? (to neglect those terms?)

- page7, lines20-21: - "This makes the inclusion of the turbulent Hall electric field Eq. (4) in the general case difficult (not speaking about the additional effect introduced by the Hall term)." —> This sentence is hard to understand. What does it mean? In what sense the inclusion of that terms is "difficult" and what is the relation with the phrase in parenthesis? ("additional effect" with respect to what?)

- page8, line19: "This assumption corresponds to the complete neglect of the turbulent dispersion relation" —> what would be the difference with the inclusion of the "turbulent dispersion relation". I am missing the link because the concept of turbulent dispersion relation is non-standard or widely known.

- page8, line28: "indicate either the presence of new waves" —> what does "new waves" mean? New with respect to what? Most of the standard theory of plasma waves is known since the 60's (Stix, etc) so I am confused about the adjective "new" here.

- page9, line5: "related to the turbulent velocity fluctuations whose spectrum under weak conditions" —-> what is exactly "weak conditions" does it mean "weak turbulence conditions". Or some other assumption that is relaxed ?

-page10, line3: Please define BMSW (I think it is an instrument on board a spacecraft different from WIND, which is not mentioned here).

-page11, line 9-10: "(We may note at this place, that concerning IK spectra it seems

improbable that they would be realised in the scale range of the ion-inertial domain.)"
–> Why? why is that improbable? Please clarify?

-page11, lines9-10: it is mentioned that the occasional presence of a bump in the turbulent density power spectra can be due to the transient features in the solar wind, or local conditions such as vthi>VA. But it would be good to also discuss related possibilities. For example, the last inequality is related to the dependence of the spectral break on the ion plasma beta (already discussed in some of the references of this manuscript but not mentioned here), but it could be related to the heliocentric distance of the observations since the local plasma conditions will be different (which is of course related to the plasma beta). This has also been pointed out before (see also General Comment 3)

REFERENCES:

Coles, W. A., & Harmon, J. K. (1989). Propagation observations of the solar wind near the sun. The Astrophysical Journal, 337, 1023. https://doi.org/10.1086/167173

Harmon, J. K., & Coles, W. A. (2005). Modeling radio scattering and scintillation observations of the inner solar wind using oblique Alfvén/ion cyclotron waves. Journal of Geophysical Research: Space Physics, 110(A3), 1–19. https://doi.org/10.1029/2004JA010834

Haverkorn, M., & Spangler, S. R. (2013). Plasma Diagnostics of the Interstellar Medium with Radio Astronomy. Space Science Reviews, 178(2–4), 483–511. https://doi.org/10.1007/s11214-013-0014-6

Howes, G. G., Tenbarge, J. M., & Dorland, W. (2011). A weakened cascade model for turbulence in astrophysical plasmas. Physics of Plasmas, 18(10). https://doi.org/10.1063/1.3646400

Spangler, S. R., & Sakurai, T. (1995). Radio interferometer observations of solar wind turbulence from the orbit of HELIOS to the solar corona. The Astrophysical Journal,

445, 999. https://doi.org/10.1086/175758

Armstrong, J. W., Cordes, J. M., & Rickett, B. J. (1981). Density power spectrum in the local interstellar medium. Nature, 291(5816), 561–564. https://doi.org/10.1038/291561a0
* * *

---

## Short Comment (SC1) · 4 Jan 2019

Thank you very much for your careful and thoughtful comments and suggestions which we will try to follow as far as possible.

Best wishes RT
* * *

---

## Author Comment (AC1) · 11 Feb 2019

We thank the reviewer for the extensive and elaborated mostly positive comments. General response:

However, the comments of the reviewer do not require responses but a nearly complete rewriting of the text. The reviewer demands so many fiundamental clarifications, which we believed would not be necessary as they are common knowledge, which cannot be given as short answers but require inclusion of short versions of all the background calculations. This we have extensively done. The result is a completely new version including restructuring. This makes a point-by-point response obsolete. We therefore below do not comment on those parts which have been rewritten but more on the

meaning. (The minor suggestions we have considered, most of them led to deletions of words or unnecessary sentences and have become obsolete, because they are covered in the rewritten text.) At the end of this response we attach the complete new version for the perusal of the reviewer. All changes, restructured parts, sections and additions are shown in blue colour. Below please find just a few explanations and answers.

The paper, submitted as a communication only, was intended to draw the attention to a simple missed fact in analysing turbulence data in the streaming solar wind. Contrary to what the reviewer believed, we do not develop any new theory of turbulence. We just apply gaugeinvariance in electrodynamics to turbulent fluctuations under the conditions of non-magnetic ions. That's all. We show that correct inclusion of gauge invariance (i.e. Lorentz force under collisionless conditions in an ideal gas) causes deviation from turbulent slope in given theoretical turbulence spectra (K- and IK-spectrum).

In application to observations in situ in space we find that conditions on the propagation angle must be satisfied for the effect to be observable. This may explain its rarity. The reviewer insists on two further points which have little to do with our approach: radio scintillations and kinetic Alfven waves. We have followed the advice and pointed to the former ground-based observations and theory. For this hint we thank the reviewer. Though radio scintillations as a diagnostic method do not really apply to what we were interested in, it makes indeed much sense to include them, in particular as these are observations from ground which are complementary to spacecraft, almost continuous and relatively easy to perform. We learned that they recently also detected deviations from K spectra both in the ISM and solar wind, which is of diagnostic interest in astrophysics for investigating the state of the ISM.

Concerning the latter we hesitate to include very much more. We cited the relevant papers and included some remarks on kinetic Alfven waves and their capacity of modifying the shape of the turbulent spectrum (as shown theoretically by Chandran et al and Wu et al.) It is true that in the ion-inertial-scale range kinetic Alfven waves can be

excited; their perpendicular wavenumber is precisely the inverse size of the ion inertial length. Assuming they are there makes sense, therefore. However, they require an external energy source, external to the stationary turbulence. This means that they are not inherent to turbulence but intermittent, depending on the presence of free energy source. Such sources are beams, temperature anisotropies, field-aligned and also diamagnetic currents. The available theories (Chandran, Wu, . . .) and observations simply assume KAW are there but do not show how they evolve in turbulence. This is like epicycle theory. Without identification of the source turbulence has little in common with their presence. If observationally confirmed, which requires inference on their polarisation, propagation direction, linear/nonlinear state and energy source, then they indicate the presence of one or the other external source. In all cases, however, they are not fundamental to turbulence but higher order effects which we are not interested in here. In contrast our approach is basic physics not having been taken into account yet in any observa-tion and interpretation, not a model theory. It requires physical awareness.

In the following we lengthily answer some of the questions of the reviewer, before adding the rewritten paper below:

The manuscript "On the ion-inertial range density power spectra in solar wind turbulence" by Treumann et al. describes a possible explanation for the occasional "bumps" seen in the power spectra density in solar wind measurements around the ion scale. This interpretation is based on the electric ion response due to quasi-neutrality and pressure balance with the magnetic power spectra.

The idea is interesting and worth to be published. The derivations seems to be logical from the basic equations and (sometimes) strong assumptions, but my main concern (in General Comment 1) is that the organization of the paper makes it hard to read (for me and maybe other readers), in the sense that is easy to lose focus: many times I did not understand the purpose of some paragraphs or derivations for backing up the main conclusion of this paper. I maybe wrong, but I would appreciate at least the opinion

of the authors about this issue. Obviously, we have not been sufficiently clear in our writing what the intentions of our efforts have been which the reviewer for this simple reason seems to have misunderstood. In order to clarify this, we have rewritten the Introduction, added references and, in order to help the reviewer, devoted the last three paragraphs of the Introduction to the wanted "roadmap" of the paper, briefly specifying its structure.

The reviewer complained about the sloppy structure of the paper. We restructured the paper such that already from the section/subsection titles it should become clear what its meaning is and where the path of the writing leads to.

GENERAL COMMENTS: 1) It is not clear to me (and maybe to some future readers of this manuscript) what part of all this discussion of this manuscript (mostly the equations) can be traced back to previous references and what part can be considered original as to the knowledge of the authors. It would be nice to add some references to the equations where appropriate. I understand that they propose a different interpretation for the bump in the density spectrum, but many of the equations seem (to me) to be widely known while others not that much (probably due to some special assumptions). I am not sure where to draw the line in order to say that from this point on it is mostly different from the standard theory of turbulence. It might be due to the organization of this manuscript, it is not that easy to follow, there are some digressions halfway that make difficult to see what is exactly the focus and what is the purpose of each section/paragraph toward the final goal, there is lack of "roadmap". In summary, the presentation and organization of the ideas could be improved.

To comment on the above: We do not develop and do not intend developing any "new theory of turbulence". Though the text is theoretical, it just aims on nothing else/nothing more than the mere clarification of the origin of the occasionally observed "bumps" in turbulent solar wind density-power spectra. There are no new equations! All equations are known. They are simply taken from wellknown electrodynamics of moving media, which implies accounting for the low-velocity relativistic effect in the Lorentzian electric

field, the notion of quasineutrality, and Poissons law. That's all.

These equations are used in linearized form which is appropriate when dealing with turbulent fluctuations. The spectra are obtained when squaring and taking Fourier transforms. This is all standard. There is nothing new concerning turbulence theory. The only trivially "new" equation is Poisson's law in its relation to the density fluctuations, which are expressed through the velocity spectrum. This is actually new, though in principle trivial. This means that the power spectrum of mechanical turbulence via Lorentz force and Poisson's law generates a turbulent density fluctuation.

This is the only "new" though nontrivial relation resulting from the requirement of quasineutrality at low-frequency turbulence.

Existing turbulence theory is merely used by us in the form of K or IK inertial power law spectra when applying the Poisson relation between the fluctuating electric field and the fluctuating density. This effect is physically restricted to unmagnetised ions only and therefore on scales inside the ion-inertial range.

To the pure theorist this is not of interest but is of vital interest to the observer. It may explain the origin of bumps. No reference to instability, intermittency, complicated interactions, any wave modes is needed, nor to any nonlinear wave and interaction theory, nor reference to kinetic theory. It is simple straightforward electrodynamics which so far was overlooked or missed by both observers and theorists. No modification of turbulence theory is included. Instead, the K and IK inertial-range spectra of turbulence theory are used as the given input to demonstrate the electrodynamic effect on them.

However, there is one important point of general interest. In the turbulence community there ghosts around the notion of "magnetic turbulence". This is wrong. The electromagnetic field itself never ever evolves into turbulence. Even in MHD all turbulence is mechanical, driven by mechanical motion, caused by free mechanical energy which sets the medium into motion and causes turbulent velocities. The conducting medium responds to it by generating currents which possess magnetic fields and cause a magnetic spectrum that is secondary to turbulence. In addition, high energy motions with beta > 1 may distort an ambient field.

Hence, in order to understand MHD turbulence one needs to understand the mechanical turbulence first! On the other hand, for beta « 1 like in neutron stars, the stiffness of the magnetic field can completely suppress any large and medium scale mechanical turbulence permitting only onedimensional turbulence along the field and very small scale turbulence on scales smaller than the particle gyroradii. The magnetic field by itself never ever becomes turbulent. In other words. MHD turbulence in media like the ISM and solar wind exists only for beta $\sim$1 and larger.

This insight has the other implication that turbulence in the velocity field for non-relativistic velocities is subject to the Galilei transformation and therefore, to first order in the fluctuations, indeed obeys Taylor's hypothesis. It can be transformed into the spacecraft frame. The magnetic field, instead, is Lorentz invariant and resists the naiv application of Taylor's hypothesis to turbulent magnetic spectra. This is a rather clear and simple basic physical conclusion, completely independent on any turbulence theory. It falsifies any unwarranted interpretation of Galilei-Taylor-transformed magnetic spectra. Its application to turbulent magnetic power spectra indicates pre-Maxwell-Lorentzian (i.e. pre-end-of-ninetienthcentury) lack of physical understanding.

2) The explanation of the flattening or bump in the density spectra of turbulent fluctuations due to Kinetic Alfvén waves is very popular (see, e.g, [Howes2011] and later reviews by the same author, [Harmon2005], etc). Why is not properly discussed here?. It is fine to put forward an additional explanation, but I do not find fair to neglect a more common one without good arguments or comparison of their respective advantages/disadvantages (KAWs are mentioned only once in this paragraph in page8, "to propagate in the kinetic Alfvén mode, or attributed to dissipation. Though this is neither impossible nor can it be excluded here, the rather more convincing conclusion is that we are dealing with de-magnetised ions in the ion-inertial domain which respond to the turbulent induction electric field and become swept over the spacecraft by the fast solar wind flow" ...and that explanations given there is not actually convincing. And even more so taking into account the large scatter in the data. ) We thank the reviewer for pointing out those references on kinetic Alfvén waves. We knew some of the papers on this item but did not feel that they deserved more than a brief hint on their existence. We did not want to ignore them but felt that they have nothing in common with our approach, as should have become clear from the above response to the general comments of the Reviewer.

However, in the revised version of the MS we do refer to them in some greater length. In the light of our above response it should become clear that our interpretation is fundamental electrodynamics, while the notion of kinetic Alfvén waves though possible, in particular in those cases where the scales suppress the relativistic density effect (see our text) supposes the working of some instability mechanism in turbulence and thus is not fundamental but second order. It also supposes the existence of a source of free energy which is not in agreement with K or IK turbulence theory where the inertial range is assumed stationary, allowing only energy flow from large to small scales, an assumption which is preserved in our basic approach when referring to K and IK velocity spectra. Thus, kinetic Alfvén waves, if confirmed, which requires additional observations of polarisation and spectral anisotropy parallel and perpendicular to the mean field, explicitly violate the inertial range turbulence theory indicating intermittency, i.e. presence of eigenmodes superimposed on the turbulence, and non-stationarity (because the highly nonlinear stationary saturation state of kinetic Alfvén waves is badly known on the background of turbulence). We have included some references and some brief dicussion of this point. We are, however, not interested in this context in any further elaboration on kinetic Alfvén waves nor in extension to inclusion of the Hall effect (which probably can be understood as an energy source for driving kinetic Alfvén waves unstable).

3) In-situ measurements are not the only way to get density power spectra in the solar wind. Scintillation observations of radio wave propagation can also provide complementary insights and cross-comparisons. That technique is often used to get density spectra in the interstellar medium ISM (mostly through diagnostics of pulsar radioemission), since scintillation is dominated by small-scale density fluctuations [Armstrong1981]. Interestingly, scintillation observations in the solar wind have also revealed a kind of "bump" (or at least a flattening) in the power density spectra at ion scales [Coles1989, Spangler1995, Harmon2005], while their counterpart in the ISM seems to be missing [Haverkorn2013 and references therein]. It would be interesting to add at least a comment about this in this manuscript.

Again, we thank the reviewer for these remarks. We have informed ourselves about the content of those papers and searched the literature for more and earlier ones. It is completely justified to mention these efforts in the context of our work even though there is no direct relation between them and our approach. This led us to include them into the reference list and mention them in the introduction in the rewritten text in the form that radio scintillation techniques (originally developed for probing the interstellar medium) early on pointed on the presence of turbulent density fluctuation spectra exhibiting even some flattening in the solar wind.

SPECIFIC COMMENTS: - page2, lines3-4: "....K or I-K spectra respectively their anisotropic generalisation"—-> This is not clear, is there may be a missing word? (respectively with respect to what?) done.

- page2, line9: Please define the angular brackets in Eq 1 (those are mentioned much later). done.

- page2, line30: What is "some unknown strongly damped virtual evanescent mode..."? A wave mode with all those adjectives is very vague, it can mean anything and confuse readers (it confused me, specially with the phrase after that says "...contribute to both density and temperature"). Please be more specific and/or provide references. deleted as not necessary.

- page2, line31: "in the higher frequency range their existence"—> please specify to

what is being referred by "their" whole sentence deleted as it is not necessary

- page3, Fig1, and page6, Figure2. How do exactly the data of those figures was obtained from the data of Safrankova2016? Please provide more details for the sake of clarity. - page3, line1: "Above frequencies > 10Ëȩ Hz.....those frequencies exceeding the ion cyclotron frequency"—-> Please specify the local ion cyclotron frequency for those in-situ measurements (otherwise it is hard to compare frequencies in Hertz with characteristic plasma frequencies in dimensionless units). done in caption and text. added all physical external conditions

- page4, lines4-5: "...electromotive force terms..... do not vary and, in considering the effects of the electric field on turbulence, can be dropped"—> This does not seem straightforward for me. Could you please be more clear on how that conclusion was reached? (to neglect those terms?)

more explicit derivation given in footnote. Electromotive forces are the space-time averaged nonlinear contributions of fluctuations (turbulence) to the mean field equations. They do not vary on the scale of the mean field equations. In the fluctuation field equations they are constants which affect these fields only through boundary conditions. In any practically infinite system like the solar wind (not accounting for its radial variation) play no role. (If radial dependences of the solar wind must be taken into account these terms could not be neglected. This is the case when comparing spectra a say 0.5 AU, 1 AU and several AU. However, for a stationary spacecraft they play no role.)

- page7, lines20-21: - "This makes the inclusion of the turbulent Hall electric field Eq. (4) in the general case difficult (not speaking about the additional effect introduced by the Hall term)." –> This sentence is hard to understand. What does it mean? In what sense the inclusion of that terms is "difficult" and what is the relation with the phrase in parenthesis? ("additional effect" with respect to what?)

OK, we thought this would be clear from the structure of the Hall term and its contribution to Poisson's equation.

But to repeat: the Hall term includes magnetic fluctuations. This is clear from the equation for the fluctuating electric field and Poisson's law. Magnetic fluctuations (by the above explanation) are secondary to turbulence, thus they are higher order. To lowest order theory the Hall contribution in Poisson's law can be neglected. Retaining them introduces a complicated higher order term in Poisson's equation for the density fluctuation. In addition, since we restricted to non-compressive turbulence, the compressive component of the magnetic fluctuation field, which is related to the Hall term, separates out. In a theory which accounts for compressibility the parallel component of the magnetic fluctuation field must be included (for instance in magnetosonic turbulence or kinetic Alfvén turbulence). Thus inclusion of the Hall term causes higher order complications. Since we separated it out, its effect requires an own separate investigation. We may, however, note that such an investigation would be related to the inclusion of kinetic Alfvén waves which the Hall term probably feeds as energy source. In this paper which deals only with fundamental electrodynamic effects we are not interested in this. One expects that in the ion-inertial (Hall) range of the K or IK (mechanical velocity turbulence) inertial spectra the mechanical energy flow from small to large wavenumbers when feeding energy into kinetic Alfvén waves should cause a steeper slope of the turbulent power density in the velocity by loosing mechanical energy. The bumps seen in the magnetic field, which in the literature are attributed to kinetic Alfvén waves, should then come up for the loss in mechanical energy and say something about the nonlinear stabilisation of the waves. We clarified this in the text.

- page8, line19: "This assumption corresponds to the complete neglect of the turbulent dispersion relation" —> what would be the difference with the inclusion of the "turbulent dispersion relation". I am missing the link because the concept of turbulent dispersion relation is non-standard or widely known.

deleted, footnote added for clarification, otherwise used in the new text in more explicit form. We structured this part into an own subsection to separate it from the main text. However, it should have become clear that we need to refer to a turbulent dispersion relation, because we use the notion of a turbulent frequency when applying the advected K and IK spectra. (We believe, advection has become clear from all the above discussion!) Advection naturally introduces a frequency at every fixed k causing Doppler broadening known since the 70th (Tennekes1975 ...). Hence at every fixed k there is a frequency \omega(k) or k(\omega) which is a dispersion relation albeit just one part of a turbulent dispersion relation.

The whole story almost always ignored in any turbulence theory is that turbulence not only proceeds in k space but also in frequency space. There would be no turbulence if every wavenumber would be independent of time respectively frequency. This was already known implicitly to Leonardo who attempted drawing of turbulence and found that it changed from one instant to the next. But this is a long story to explain. To make it brief: Stationary turbulence (Batchelor 1950 for instance) is stationary only in the time average. Sitting on one eddy (fixed k) one sees the eddy grow, circulate and decay. It depends on time. Thus any complete description even of stationary turbulence implies a Fourier transform in k and \omega. This spans a volume in 4-d space, one axis frequency. This turbulent volume or surface is a multivalued turbulent dispersion relation. Of this, turbulence considers just the complete integral over frequency, thus a pure k dependence remains. The dispersion relation is integrated out. When of course assuming Alfvenic turbulence, then one implicitly imposes a relation between some ks and frequency \omega which theoretically simplifies the problem. But in practice such observations proceed in time and must be separately justified as providing k spectra, which is a subtle problem and, at least to our knowledge, causes a problem the solution of which is no simple matter. Even multi-spacecraft missions are unable to measure the k spectrum because of the roughness of spacecraft separation. One needs a multitude of irregularly floating spacecraft, a cloud, to measure an approximate local spectrum, and one needs a sophisticated technique of analysing those measurements. Also groundbased scintillation observations can, at least to our knowledge, so far not provide a good k spectrum. Possibly one would need a worldwide network with many scales on the earth for this purpose.

We here are dealing with observations of density power spectra in time. We assume that they are caused by turbulence in velocity. For this we assume K or IK spectra (tentatively believing in the validity of K or IK theory of turbulence which integrate over frequency). Of the velocity power spectra we know that in a fast streaming flow they map via Taylors assumption from k into \omega space of a fixed observer (spacecraft). Hence, the turbulent dispersion relation has been averaged out via K and IK assumptions. What possibly remains is the advective deformation through Doppler broadening which causes a reintroduction of a frequency into K and IK k-spectra which however is replaced through the known Doppler dependence, causing a slight deformation of the K and IK spectra. So this is a long and complicated story of a simple fact (that turbulence is by no means only in space!) which nobody wants to read and nobody wants to deal with. Turbulence theory makes it easy by integrating over \omega. To reconcile this with observations is no simple matter. Taylor's assumption, in particular, does not hold for magnetic power spectra. But this is another complicated story.

- page8, line28: "indicate either the presence of new waves" —> what does "new waves" mean? New with respect to what? Most of the standard theory of plasma waves is known since the 60's (Stix, etc) so I am confused about the adjective "new" here.

Deleted. We do not refer to plasma waves, principally, as they have nothing im common with turbulence. However, we note that Stix only includes eigenmodes which are at best weakly damped. These are the usual kinetic plasma waves. There is a large number of strongly damped waves which are not solutions of the weakly damped linear dispersion relation, not speaking about nonlinear waves. Stix calls these evanescent (mentioning a few of them). They all have damping rates exceeding the frequency. In turbulence all these waves can be populated for the time the energy passes across their k range, i.e. the time needed to jump from one k to another nearby k. If this time is shorter than the damping time, the modes are present in the spectrum, not being real but virtual waves. In the spectrum they contribute to the slope. If the spectrum could be resolved down

to fractions of k one would also detect holes in it where the time of passage for the energy does not allow population of the mode as the mode dampes away faster. The fluctuations seen building up the detected spectrum (at the observational resolution) are simply Fourier transformed by the instrument or programme which constructs a continuous Fourier spectrum of a certain shape. However this spectrum contains all kinds of waves: eigenmodes, sidebands, those evanescent modes and all kinds of nonlinearly scattered modes. This is what was meant. Turbulence theorists are well aware of all these contributions which cannot be resolved.

- page9, line5: "related to the turbulent velocity fluctuations whose spectrum under weak conditions" —-> what is exactly "weak conditions" does it mean "weak turbulence conditions". Or some other assumption that is relaxed ?

No, it does not mean weak turbulence. Weak turbulence means a sequence of eigenmodes which interact in such a way that a small expansion parameter (energy ratio) exists which can be used to organise the different interactions into a hierarchy of two, three, four and so on wave collisions with coupling coefficients. These interactions are based on a hierarchy of resonances. Weak turbulence does not apply to MHD turbulence. It fails badly as it cannot explain any measured spectrum because real turbulence is not built from resonant interactions. This criticism applies to the proposal in the literature that "turbulence is built up from collisions between Alfven waves". This is wrong. It is old-fashioned thinking of the time when nothing was known of phase transitions and virtual modes. In principle turbulence is a sequence of phase transitions involving virtual modes which in the K inertial range form a continuum. Just because this is so and cannot be treated hierarchically, the K range exists. We have reworded the text. Weak conditions here means just our approximations and their limits. So we wrote: "In the limits of our approximations."

-page10, line3: Please define BMSW (I think it is an instrument on board a spacecraft different from WIND, which is not mentioned here). Done, thanks.

-page11, line 9-10: "(We may note at this place, that concerning IK spectra it seems C4 ANGEOD Interactive comment Printer-friendly version Discussion paper improbable that they would be realised in the scale range of the ion-inertial domain.)" –> Why? why is that improbable? Please clarify? Done, respectively deleted. Not important.

-page11, lines9-10: it is mentioned that the occasional presence of a bump in the turbulent density power spectra can be due to the transient features in the solar wind, or local conditions such as vthi>VA. But it would be good to also discuss related possibilities. For example, the last inequality is related to the dependence of the spectral break on the ion plasma beta (already discussed in some of the references of this manuscript but not mentioned here), but it could be related to the heliocentric distance of the observations since the local plasma conditions will be different (which is of course related to the plasma beta). This has also been pointed out before (see also General Comment 3)

All deleted as not necessary. But we have given a precise discussion of the range and limitations and in application to the K and IK spectra have made extensive use. Once more many thanks to the Reviewer. Please find below the new rewritten and substantially extended text in view of your comments.

Like here, all new text and equations are in blue colour.

Please also note the supplement to this comment:
https://www.ann-geophys-discuss.net/angeo-2018-129/angeo-2018-129-AC1-supplement.pdf

**Supplement:**

[revised manuscript text omitted]

$$r_q \sim 0.24\beta_{iq} \quad \text{AU} \tag{46}$$

15   which corresponds to the region inside $< 51\beta_{iq}$ R$_\odot$ from the sun. The plasma-$\beta$ at the source region is not known. It requires the assumption of a model. Hence, up to its determination we know the distance of the source. Since it must lie inside 1 AU, we also conclude that $\beta_{iq} \lesssim 4.5$. This number is not unreasonable though it might be too large, as one would expect that $\beta_{iq} \lesssim 1$ (McKenzie et al., 1995).

**5.7   Summary and outlook**

20   We considered the cases $V_0 = U_0 = 0$, $V_0 = 0$, $U_0 \neq 0$, and $V_0 \neq 0$ for K and IK velocity spectra. The resulting spectral slopes are given as $b$ in Table 1 fourth column. The input spectral power densities are $\mathcal{E}_{IK}, \mathcal{E}_{IK}^{ad}$. Each yields a different ion inertial scale range power spectrum in $k$-space and, consequently, also a different power law spectrum in $\omega_s$-space.

Table 1 shows that the ordinary spectra acquire positive slopes in wave number $k$ in the frame of stationary turbulence in the turbulence frame. However, observations of this slope in frequency undermine this conclusion, suggesting that it is the ordinary

25   K (IK) velocity turbulence (or if anisotropy is taken into account, the Kolmogorov-Goldreich-Sridhar KGS) spectrum in the ion-inertial range which then deforms the density spectrum. All advected spectra have negative slope in frequency which in this form disagrees with observation. The obtained advected slopes in the stationary turbulence frame are too far away from the flattest negative slope of $\omega_s^{-1}$ for being related. This implies that spacecraft observations interpreted as observing the local stationary turbulence do not see an advected spectrum in the ion-inertial K (IK) inertial range. Their nominal K and IK slopes

30   are $-\frac{2}{3}$ and $-\frac{1}{2}$ respectively, still too far away from the observe $\omega^{-1}$ spectral slope. They are, however, capable of explaining the high-frequency spectral bump in the WIND spectrum given in Fig. **??** which has the correct advective IK spectral index $-\frac{1}{2}$ convected across the WIND spacecraft before spectral decay.

However, positive slopes or bumps are observed rarely. The more frequent case is flattened negative slopes. Such slopes are suggested when advected spectra are convected by the fast solar wind across the spacecraft. Their nominal K and IK slopes are $-\frac{2}{3}$ and $-\frac{1}{2}$ respectively, still too far away from the observe $\omega^{-1}$ spectral slope. They are, however, capable of explaining the high-frequency spectral bump in the WIND spectrum given in Fig. 5 which has the correct advective IK spectral index $-\frac{1}{2}$ convected across the WIND spacecraft before spectral decay.

Generally the form of a distorted power spectrum in density depends on the external solar wind conditions. The reconciliation of these with the theoretical predictions and the observation of the spectral range of the distortion is a difficult task. We have attempted it in the discussion section above but cannot present any undisputable results as these require a more sophisticated investigation both in theory and based on better resolutions than those which have been available to us. In particular the bending of the power spectral density to lower frequencies requires further inquiry into theory and observation in order to identify the maximum point of the advected spectrum in frequency and glue the spectrum to the undisturbed K or IK inertial ranges.

We also note that so far we have not taken into account the contribution of Hall spectra. These affect the shape of the density spectrum via the Hall magnetic field, a second order effect indeed which, however, might contribute to spectral deformation. These Hall fields are closely related to kinetic effects in the ion-inertial range which might cause the excitation of kinetic Alfvén waves which are believed to also contribute to deformations of the I and IK inertial range spectra in the ion-inertial region. We mentioned this possibility in the introduction where we pointed to the relevant publications.

Hall magnetic fields will necessarily be present on those scales for the simple reason that electrons and ions in an advected and/or convected magnetised plasma perform independent motions which generate Hall currents to flow across the magnetic field. These become internal turbulence generated free energy sources which could drive kinetic Alfvén waves unstable. The effect is second order when compared with the zero order Lorentz invariance to that we have referred. Nevertheless, since the perpendicular scales $k_\perp \sim \lambda_i^{-1}$ for kinetic Alfvén waves do perfectly agree with the ion-inertial range their presence seems quite natural.

Similarly, small-scale shock waves might evolve at the inferred high Mach numbers when turbulent eddies grow and steepen in the small scale range. These necessarily become sources of electron beams, reflect ions, and transfer their energy in a kinetic-turbulent way to the particle population. Such beams then act as sources of particular wave populations which contribute to turbulence preferably at kinetic scales. Inclusion of all these effects is a difficult task. It still opens a wide field for investigation of turbulence at the kinetic scales on the ion-inertial scale.

[revised manuscript text omitted]

---

## Author Response (AR1)

We thank the reviewer for the extensive and elaborated mostly positive comments.

General response:

However, the comments of the reviewer do not require responses but a nearly complete rewriting of the text. The reviewer demands so many fiundamental clarifications, which we initially believed that they would not have been necessary as they are mostly common knowledge. These crarifications cannot be replied to as short answers but require inclusion of short versions of some background calculations.

A.
This we have extensively done (see footnote 2, Sections 2.4-2.5, 3.2-3.3, 5.2-5.6). The result is a rather complete new version including a completely new restructuring of the MS into sections and subsections with headings making clear what they contain.

In principle, this makes a point-by-point response to the revewer's points obsolete. We therefore below do not comment on those parts which have been rewritten but more on the meaning. (The minor suggestions we consider separately. Most of them led to deletions of words or unnecessary sentences and have become obsolete as well, because they are covered in the rewritten text or dropped.)

All changes, restructured parts, sections and additions are in the resubmitted manuscript shown in blue colour. They should pop out.

Below please find just a few (though lengthy) explanations and answers. The reviewer might be interested in them. Otherwise he can skip them and go directly to the manuscript. We, however, suggest reading tem, which will take a few minutes.

B.
The paper, originally submitted as a communication only, was intended to draw the attention to a rather simple missed fact in analysing turbulence data in the streaming solar wind. Contrary to what the reviewer believed, we do not develop any new theory of turbulence. We just apply gauge-invariance in electrodynamics to turbulent fluctuations under the conditions of non-magnetic ions. That's all. We show that correct inclusion of gauge invariance (i.e. Lorentz force under collisionless conditions in an ideal conducting gas = plasma) causes deviation from the turbulent power-spectral slope in given theoretical turbulence spectra (K- and IK-spectrum).

In application to observations in situ space we find that conditions on the propagation angle and ion-to-electron-temperature ratio (Fig.2) must be satisfied for the effect to be observable. This may explain its rarity.

C.
The reviewer insists on two further points which have little to do with our approach: radio scintillations and kinetic Alfven waves.

We have followed the advice and pointed to the former ground-based observations and theory(Sections 1 & 5). For this particular hint we are very grateful. We did not think about scintillations so far but find them indeed appealing as being ground based.

Though radio scintillations as a diagnostic method do not really apply to what we were interested in, it makes indeed much sense to include them, in particular as these are observations from ground which are complementary to spacecraft, almost continuous and relatively easy to perform. We learned that they recently also detected deviations from K spectra both in the ISM and solar wind.

Concerning the latter we hesitate to include very much more. We cited the relevant papers and included some remarks on kinetic Alfven waves and their capacity of modifying the shape of the turbulent spectrum (as shown theoretically by Chandran et al and Wu et al.)

What concerns kAW, it is true that in the ion-inertial scale range kAW can propagate. Their perpendicular wavenumber is precisely the inverse ion inertial length. Hence these scales support them as eigenmodes of the plasma on those scales. However, for excitation they require an external energy source, external to the stationary turbulence.

This means that when excited they are not inherent to turbulence but intermittent, depending on the presence of free energy source. Such sources are beams, temperature anisotropies, field-aligned and also diamagnetic currents which are not components of homogeneous stationary turbulence. The available theories (Chandran, Wu, …) and observations simply assume KAW are there but do not show how they evolve in homogeneous stationary turbulence. This is rather different from our philosophy whichc deals solely with homogeneous isotropic inertial range turbulence.

Without identification of the source of kAWs turbulence has little in common with their presence. If observationally confirmed, which requires inference on their polarisation, propagation direction, linear/nonlinear state and energy source, then they indicate the presence of one or the other external source. In all cases, however, they are not fundamental to turbulence but higher order effects which we are not interested in here.

In contrast, our approach is basic, independent of such sources or intermittency. So far these ideas have not been taken into account yet in any observation and interpretation. Our approach is not a model theory of turbulence. It simply refers to fundamental electrodynamics.

In the following we lengthily answer some of the questions of the reviewer:

0. The manuscript "On the ion-inertial range density power spectra in solar wind turbulence" by Treumann et al. describes a possible explanation for the occasional "bumps" seen in the power spectra density in solar wind measurements around the ion scale. This interpretation is based on the electric ion response due to quasi-neutrality and pressure balance with the magnetic power spectra.

The idea is interesting and worth to be published. The derivations seems to be logical from the basic equations and (sometimes) strong assumptions, but my main concern (in General Comment 1) is that the organization of the paper makes it hard to read (for me and maybe other readers), in the sense that is easy to lose focus: many times I did not understand the purpose of some paragraphs or derivations for backing up the main conclusion of this paper. I

maybe wrong, but I would appreciate at least the opinion of the authors about this issue.

Reply to 0.
Thanks very much for these positive comments.
However, obviously we have not been sufficiently clear in writing what the intention of our efforts have been. For this simple reason the reviewer seems to have misunderstood it.

In order to clarify what we are aiming for, we have rewritten the Introduction, added references (including those the Reviewer suggested and others) and, in order to help the reviewer, devoted the last three paragraphs of the Introduction to the wanted "roadmap" of the paper, briefly specifying its structure.

The reviewer complained about the sloppy structure of the paper. According to his demand, we restrutured the paper such that already from the section/subsection titles it should become clear what the meaning of the sections is and where the path leads to.

GENERAL COMMENTS:

1) It is not clear to me (and maybe to some future readers of this manuscript) what part of all this discussion of this manuscript (mostly the equations) can be traced back to previous references and what part can be considered original as to the knowledge of the authors. It would be nice to add some references to the equations where appropriate. I understand that they propose a different interpretation for the bump in the density spectrum, but many of the equations seem (to me) to be widely known while others not that much (probably due to some special assumptions). I am not sure where to draw the line in order to say that from this point on it is mostly different from the standard theory of turbulence. It might be due to the organization of this manuscript, it is not that easy to follow, there are some digressions halfway that make difficult to see what is exactly the focus and what is the purpose of each section/paragraph toward the final goal, there is lack of "roadmap". In summary, the presentation and organization of the ideas could be improved.

Reply to 1.

We do not develop and do not intend developing any "new theory of turbulence". Though the text is 50% theoretical, it just aims on nothing else/more than the mere clarification of the origin of the occasionally observed "bumps" in turbulent solar wind density-power spectra. This requires no theory like kAWs or nonlinear scatterings, interactions.

There are no new equations! All equations are known. They are familiar in the electrodynamics of moving media, which implies accounting for the low-velocity relativistic effect in the Lorentzian electric field (i.e. the gauge invariance), the natural notion of quasineutrality in application to low frequency turbulence, and Poissons law. That's all.

These equations are used in linearized form which is appropriate when dealing with fluctuations. The spectra are obtained when squaring and taking Fourier transforms. This is all standard. There is nothing new concerning turbulence theory. The only trivially "new" equation is Poisson's law in its relation to the density fluctuations, which are expressed through the velocity spectrum. Its use is actually new, though in principle trivial. This means that the power spectrum of mechanical turbulence via Lorentz force and Poisson's law

generates turbulent density fluctuations in order to restore quasineutrality.

This is the only "new" and nontrivial relation resulting from the necessary natural requirement of quasineutrality in low-frequency turbulence which otherwise would lead to charge accumulations.

Existing turbulence theory is merely quoted by us in the form of K or IK inertial power law spectra when ultimately applying the Poisson relation between the fluctuating electric field and the fluctuating density. This effect is restricted to unmagnetised ions only and therefore appears merely on scales inside the ion-inertial range where electrons remain magnetised.

To the pure theorist this is not of much interest but is of vital interest to the observer. It may explain the origin of bumps, at least it contributes to their explanation. No reference to instability, intermittency, complicated interactions, any wave modes is needed, nor to any nonlinear wave and interaction theory, nor reference to kinetic theory. It is simple straightforward electrodynamics which so far was overlooked or missed by both observers and theorists. No modification of turbulence theory is included. Instead, the K and IK inertial range spectra of turbulence theory are used as given input to demonstrate the electrodynamic effect on them.

However, there is one important point of general interest. In the turbulence community there ghosts around the notion of "magnetic turbulence". This is wrong. The electromagnetic field itself never ever evolves into turbulence. Even in MHD all turbulence is mechanical, driven by mechanical motion, caused by free mechanical energy which sets the medium into motion and causes turbulent velocities. The conducting medium responds to it by generating currents which possess magnetic fields and cause a magnetic spectrum that is secondary to turbulence. In addition, high energy motions with beta > 1 may distort an ambient field.

Hence, in order to understand MHD turbulence one needs to understand the mechanical turbulence first! This also applies to the ISM.

On the other hand, for beta << 1 like in neutron stars, the stiffness of the external magnetic field can completely suppress any large and medium scale mechanical turbulence permitting only one-dimensional turbulence along the field and very small-scale perpendicular turbulence on scales smaller than the particle gyroradii. The magnetic field by itself never ever becomes turbulent. In other words, MHD turbulence in weak-magnetic field media like the ISM and solar wind is most important at beta ~1 and larger.

This insight has the other implication that turbulence in the velocity field for non-relativistic velocities is subject to the Galilei transformation and therefore, to first order in the fluctuations, indeed obeys Taylor's hypothesis. It can be transformed into the spacecraft frame. The magnetic field, instead, is Lorentz invariant and resists the naiv application of Taylor's hypothesis to turbulent magnetic spectra. This is a rather clear and simple basic physical conclusion, completely independent on any turbulence theory. It falsifies any unwarranted interpretation of Galilei-Taylor-transformed magnetic spectra.

2) The explanation of the flattening or bump in the density spectra of turbulent fluctuations due to Kinetic Alfvén waves is very popular (see, e.g, [Howes2011] and later reviews by the same author, [Harmon2005], etc). Why is not properly discussed here?. It is fine to put

forward an additional explanation, but I do not find fair to neglect a more common one without good arguments or comparison of their respective advantages/disadvantages (KAWs are mentioned only once in this paragraph in page8, "to propagate in the kinetic Alfvén mode, or attributed to dissipation. Though this is neither impossible nor can it be excluded here, the rather more convincing conclusion is that we are dealing with de-magnetised ions in the ion-inertial domain which respond to the turbulent induction electric field and become swept over the spacecraft by the fast solar wind flow" ...and that explanations given there is not actually convincing. And even more so taking into account the large scatter in the data. )

Reply to 2:
We thank the reviewer for pointing out those references on kinetic Alfvén waves. We knew some of the papers on this item but did not feel that they deserved more than a brief hint on their existence. We did not want to ignore them but felt that they have nothing in common with our approach, as should have become clear from the  above response to the general comments of the Reviewer.

However, in the revised version of the MS we do refer to them in some length. In the light of our above response it should become clear that our interpretation is fundamental electrody-namics, while the notion of kinetic Alfvén waves though doubtlessly possible, in particular in those cases where the scales suppress the relativistic density effect (see our text) supposes the working of some instability mechanism in turbulence and thus is not fundamental but second order.

kAWs also suppose the existence of a source of free energy, which is not in agreement with K or IK turbulence theory where the inertial range is assumed stationary, and the energy source is at very large wavelengths resp. very small wavenumbers. It allows only energy flow from large to small scales, an assumption which is preserved in our basic approach when referring to K and IK velocity spectra.

Kinetic Alfvén waves, if confirmed, which requires additional observations of polarisation, spectral anisotropy parallel and perpendicular to the mean field, explicitly violate the inertial range turbulence theory, indicating intermittency, i.e. presence of eigenmodes superimposed on the turbulence, and non-stationarity (because the highly nonlinear stationary saturation state of kinetic Alfvén waves is badly known on the background of turbulence). We have included the suggested and some more references and added some brief discussion of this point. We are, however, not interested in this context in any further elaboration on kinetic Alfvén waves nor in an extension to inclusion of the Hall effect (which is typical for the ion inertial range and might be understood as an energy source for driving kinetic Alfvén waves unstable but itself is a second order effect carrying little energy which hardly suffices to generate a bulge on the spectrum).

3) In-situ measurements are not the only way to get density power spectra in the solar wind. Scintillation observations of radio wave propagation can also provide complementary insights and cross-comparisons. That technique is often used to get density spectra in the interstellar medium ISM (mostly through diagnostics of pulsar radioemission), since scintillation is dominated by small-scale density fluctuations [Armstrong1981]. Interestingly, scintillation observations in the solar wind have also revealed a kind of "bump" (or at least a flattening) in the power density spectra at ion scales [Coles1989, Spangler1995, Harmon2005], while their counterpart in the ISM seems to be missing [Haverkorn2013 and

references therein]. It would be interesting to add at least a comment about this in this manuscript.

Rely to 3:
This is indeed an important point. We thank the reviewer for these remarks.

We have informed ourselves about the content of those papers and searched the literature for more and earlier ones. It is completely justified to mention these efforts in the context of our work even though there is no direct relation between them and our approach. This led us to include them into the reference list and mention them in the introduction and briefly in the discussion in the rewritten text in the form that radio scintillation techniques (originally developed for probing the interstellar medium) early on pointed on the presence of turbulent density fluctuation spectra exhibiting even some flattening in the solar wind respectively deviation from the spectral slope. In particular the diagnostic aspect of scintillations is rather interesting as they are groundbased and can with varying baselines relatively easily and continuously performed.

SPECIFIC COMMENTS:

- page2, lines3-4: "....K or I-K spectra respectively their anisotropic generalisation"—-> This is not clear, is there may be a missing word? (respectively with respect to what?)

done.

- page2, line9: Please define the angular brackets in Eq 1 (those are mentioned much later).

done.

- page2, line30: What is "some unknown strongly damped virtual evanescent mode..."? A wave mode with all those adjectives is very vague, it can mean anything and confuse readers (it confused me, specially with the phrase after that says "...contribute to both density and temperature"). Please be more specific and/or provide references.

deleted as not necessary in this context.

- page2, line31: "in the higher frequency range their existence"—> please specify to what is being referred by "their"

whole sentence deleted as it is not necessary

- page3, Fig1, and page6, Figure2. How do exactly the data of those figures was obtained from the data of Safrankova2016? Please provide more details for the sake of clarity. - page3, line1: "Above frequencies > 10^ Hz.....those frequencies exceeding the ion cyclotron frequency"—-> Please specify the local ion cyclotron frequency for those in-situ measurements (otherwise it is hard to compare frequencies in Hertz with characteristic plasma frequencies in dimensionless units).

done in caption and text. added all available physical external conditions in the solar wind.

- page4, lines4-5: "...electromotive force terms..... do not vary and, in considering the effects of the electric field on turbulence, can be dropped"—> This does not seem straightforward for me. Could you please be more clear on how that conclusion was reached? (to neglect those terms?)

more explicit derivation given in footnote 2. Electromotive forces are the space-time averaged nonlinear contributions of fluctuations (turbulence) to the main field equations. They do not vary on the scale of the mean field equations. In the fluctuation field equations they are constants which affect these fields only through boundary conditions. In any practically infinite system like the solar wind (not accounting for its radial variation) play no role. (If radial dependences of the solar wind must be taken into account these terms could not be neglected. This is the case when comparing spectra a say 0.5 AU, 1 AU and several AU. However, for a stationary spacecraft they play no role.)

- page7, lines20-21: - "This makes the inclusion of the turbulent Hall electric field Eq. (4) in the general case difficult (not speaking about the additional effect introduced by the Hall term)." –> This sentence is hard to understand. What does it mean? In what sense the inclusion of that terms is "difficult" and what is the relation with the phrase in parenthesis? ("additional effect" with respect to what?)

OK, we thought this would be clear from the structure of the Hall term and its contribution to Poisson's equation.

The Hall term includes magnetic fluctuations. This is clear from the equation for the fluctuating electric field and Poisson's law. Magnetic fluctuations (by the above explanation) are secondary to turbulence, thus they are higher order. To lowest order theory, the Hall contribution in Poisson's law can be neglected. Retaining them introduces a complicated higher order term in Poisson's equation for the density fluctuation. In addition, since we restricted to non-compressive turbulence, the compressive component of the magnetic fluctuation field related to the Hall term separates out. In a theory which accounts for compressibility the parallel component of the magnetic fluctuation field must be included (for instance in magnetosonic turbulence or kinetic Alfvén turbulence). Inclusion of the Hall term causes higher order complications. Since we separated it out, its effect requires an own separate investigation. We may, however, note that such an investigation would be related to the inclusion of kinetic Alfvén waves which the Hall term probably feeds as energy source (see our above commments).

In this paper which deals only with fundamental electrodynamic effects we are not interested in this. One expects that in the ion-inertial (Hall) range of the K or IK (mechanical velocity turbulence) inertial spectra the mechanical energy flow from small to large wavenumbers when feeding energy into kinetic Alfvén waves should cause a weakly steeper slope of the turbulent power density in the velocity by loosing mechanical energy. The bump seen in the magnetic field, which in the literature is attributed for instance to kinetic Alfvén waves, should then come up for the loss in mechanical energy and say something about the nonlinear stabilisation of the waves. Investigating this lies outside our intention but is certainly of interest in a theory which accounts for higher than first order effects.

We clarified this in the text.

- page8, line19: "This assumption corresponds to the complete neglect of the turbulent dispersion relation" —> what would be the difference with the inclusion of the "turbulent dispersion relation". I am missing the link because the concept of turbulent dispersion relation is non-standard or widely known.

deleted, footnote 5 added for some clarification.

We structured this part into an own subsection 3.2 to separate it from the main text.

Remark on this point:

However, it should have become clear that we need to refer to kind of a turbulent dispersion relation, because we use the notion of a turbulent frequency when appying the advected K and IK spectra. Advection naturally introduces a frequency at every fixed k causing Doppler broadening known since the 70th (Tennekes1975 …). Hence at every fixed k there is a frequency \omega(k) or k(\omega) which formally is a dispersion relation albeit  just one part of a turbulent dispersion relation.

The whole story almost always ignored in any turbulence theory is that turbulence not only proceeds in k space but also in frequency space. There would be no turbulence if every wavenumber would be independent of time respectively frequency.

This was already known implicitly to Leonardo who attempted drawing of turbulence and found that it changed from one instant to the next. To make it brief: Stationary turbulence (Batchelor 1949 for instance) is stationary only in the time average. Sitting on one eddy (fixed k) one sees the eddy grow, circulate and decay. It depends on time.

Remark on k-spectrum:

Any complete description even of stationary turbulence implies a Fourier transform in k and \omega. This spans a volume in 4-d space, one axis frequency. This turbulent surface is a multivalued turbulent dispersion relation \omega(k). Turbulence theory considers just the integral over frequency, thus a pure k dependence remains. The dispersion relation is integrated out. When assuming Alfvenic turbulence, then one implicitly imposes a linear relation between k and frequency \omega which theoretically simplifies the problem. But in practice such observations proceed in time and must be separately justified as providing k spectra, which is a subtle observational problem and, at least to our knowledge, causes a problem the solution of which is no simple matter. Even multi-spacecraft missions are unable to measure the k spectrum sufficiently precisely because of the roughness of spacecraft separation. One needs a multitude of irregularly flowing spacecraft, a cloud, to measure an approximate local spectrum, and one needs a sophisticated technique of analysing those measurements. Groundbased scintillation observations can, at least to our knowledge, so far not provide a good k spectrum. Possibly one would need a worldwide network with many scales on the earth for this purpose.

We here are dealing with observations of density power spectra in time. We assume that they are caused by turbulence in velocity. For the letter we refer to K or IK spectra (tentatively believing in the validity of K or IK theory of turbulence). Of the velocity power  spectra we know that in a fast streaming flow they map via Taylors assumption k into \omega space of a

fixed observer (spacecraft). Hence, the turbulent dispersion relation has been averaged out via K and IK assumptions. What possibly remains is the advective deformation through Doppler broadening which causes a re-introduction of a frequency into K and IK k-spectra which however is replaced through the known Doppler dependence, causing a slight deformation of the K and IK spectra.

This is a long and complicated story of a simple fact (that turbulence is by no means only in space!) which nobody wants to read and nobody wants to deal with. Turbulence theory makes it easy by integrating over \omega. To reconcile this with observations is no simple matter. Taylor's assumption, in particular, does not hold for magnetic power spectra. But this is another complicated story.

- page8, line28: "indicate either the presence of new waves" —> what does "new waves" mean? New with respect to what? Most of the standard theory of plasma waves is known since the 60's (Stix, etc) so I am confused about the adjective "new" here.

Deleted. We do not refer to plasma waves, principally, as they have nothing im common with turbulence.

However, we note that Stix only includes eigenmodes which are weakly damped. These are the usual kinetic plasma waves. In addition, there is a large number of strongly damped waves which are not solutions of the weakly damped linear dispersion relation, not speaking about nonlinear waves. Stix calls these evanescent (mentioning a few of them). They have damping rates exceeding the frequency. In turbulence these waves can be populated for the time the energy passes across their k-range, i.e. the time needed to jump from one k to another nearby k. If this time is shorter than the damping time, the modes are present in the spectrum, not being real but virtual waves. In the spectrum they contribute to the slope. If the spectrum could be resolved down to fractions of k one would also detect holes in it where the time of passage for the energy does not allow population of the mode as the mode dampes away. The fluctuations seen building up the detected spectrum (at the observational resolution) are simply Fourier transformed by the instrument or programme which constructs a continuous Fourier spectrum of a certain shape. However this spectrum contains all kinds of waves: eigenmodes, sidebands, those evanescent modes and all kinds of nonlinearly scattered modes. This is what was meant. Turbulence theorists are well aware of all these contributions which cannot be resolved.

- page9, line5: "related to the turbulent velocity fluctuations whose spectrum under weak conditions" —-> what is exactly "weak conditions" does it mean "weak turbulence conditions". Or some other assumption that is relaxed ?

No it does not mean "weak turbulence". Weak turbulence means a sequence of eigenmodes which interact in such a way that a small expansion parameter (energy ratio) exists which can be used to organise the different interactions into a hierarchy of two, three, four and so on wave collisions. These interactions are based on a hierarchy of resonances.

Weak turbulence does not apply to MHD turbulence. It fails badly as it cannot explain any measured spectrum because real turbulence is not built from resonant interactions. This critique applies to the proposal in the literature that "turbulence is built up from collisions between Alfven waves". Turbulence is mechanical. The magnetic field reacts passively to it.

Mechanical turbulence generates small-scale and large-scale currents. The magnetic fluctuations are the magnetic field of those currents. Some part of them can be described as alfven waves.

In principle turbulence is a sequence of phase transitions which in the K inertial range form a continuum. Just because this is so and cannot be treated hierarchically, the K range exists.

We have reworded the text in this spirit. Weak conditions here means just our weak approximations and their limits. So we wrote: "In the limits of our approximations."

-page10, line3: Please define BMSW (I think it is an instrument on board a spacecraft different from WIND, which is not mentioned here).

Done, thanks.

-page11, line 9-10: "(We may note at this place, that concerning IK spectra it seems improbable that they would be realised in the scale range of the ion-inertial domain.)" –> Why? why is that improbable? Please clarify?

Done, respectively deleted. Not important.

-page11, lines9-10: it is mentioned that the occasional presence of a bump in the turbulent density power spectra can be due to the transient features in the solar wind, or local conditions such as vthi>VA. But it would be good to also discuss related possibilities. For example, the last inequality is related to the dependence of the spectral break on the ion plasma beta (already discussed in some of the references of this manuscript but not mentioned here), but it could be related to the heliocentric distance of the observations since the local plasma conditions will be different (which is of course related to the plasma beta). This has also been pointed out before (see also General Comment 3)

All deleted as not necessary. But we have given a precise discussion in subsection 2.4 and the discussion section of the range and limitations in application to the K and IK spectra.

Once more many thanks to the Reviewer.

We have very much appreciated the comments and suggestions, taken them into account in rewriting the paper to make it better accessible.

We apologise for this rather lengthy reply and explanations. But we felt that his was needed, not only to explain our efforts to the reviewer but also to write down some very general points which apply to observation and theory of turbulence when including magnetic fields.

Like here, all new text and equations are in blue colour.

---

## Author Response (AR2)

Response to the reviewer:

I thank the authors for the very detailed response to all of my comments and for the very extensive rewriting of their manuscript. I sincerely was actually not expecting that. I just suggested some small clarifications to make the manuscript more understandable for a wider audience. In addition, I did not consider that they were many fundamental clarifications other than the unclear writing of some sentences (sorry if I was too pedantic with that). Other readers may have misunderstood the point of this paper in the same way as I. So, I sincerely apologize for all the additional work that my suggestions involuntarily caused to the authors.

Thanks, however there would be no reason to apologize. The responses were very useful and, as the opinion of the reviewer reflects, contributed essentially to a clearer presentation. Thanks again.

In my opinion, the revised version of the manuscript now is more clear than before.
The organization of the different sections into subsections makes everything easier to follow.
In particular, the last paragraph of page 4 (part of the introduction): I think this is one of the most important addition of the revised version of this manuscript, which explains its main point and purpose, what makes this manuscript different from other approaches explained in the previous paragraph (in page 3). The last paragraph in page 3 also motivates very well the purpose of this paper in explaining the observed bumps in the turbulent spectra. The description written in the second paragraph in page 4 is quite helpful to understand the motivation behind the idea presented in this paper. The same goes for the very helpful first paragraph in page 17 (section 5, discussion).

I have only a few small comments and possible typos to be corrected, actions items that it would be good to address before publication, please. All of them are only related with the new additions/corrections of this version of the manuscript and should not require to change more than a few words/one sentence each. This is just to try to make the manuscript more precise, please.

We appreciate this and the following comment very much indeed.
* * *
REMARKS:

1) page 4 "If kinetic Alfvén waves are unambiguously confirmed, the inner solar wind ... must be subject to the continuous presence of small scale collisionless shocks" ---> This is a strong statement, which requires

more evidence, otherwise it would be misleading. Why the "must"? I am not sure if that is the only possible choice, the presence of KAWs could be associated to shocks, but I think it is not strictly required.

Right. This is too strong an expression. Replaced by "could".

However, the whole story is more complicated. As expressed in the former replies and briefly noted at another place in the MS, thinking of solar wind turbulence in terms of homogeneous and stationary turbulence is incorrect. If one does so, then one has in mind the local state of turbulence (i.e. at the location of the spacecraft which measures), but in this case the system is open. On the larger scale radial dependence and thus evolution cannot be neglected. The source of the turbulence is somewhere in or close to the solar corona. The turbulence therefore evolves when streaming away. In the ion-inertial range one would expect that in the source region KAWs will become excigted by the presence of some free energy which causes the turbulence on all scales. Excitation is certainly much faster in time than Kolmogorov's cross-spectrum flow which hold just for stationary turbulence. The KAWs will grow, usually to small amplitudes, they possibly saturate quasilinearly. However being waves in collisionless conditions they are nonlinear kind of simple waves. This implies that during transport radially downstream some of them will become damped, other with grow and steepen on a short scale. So, if they have been exited in the corona and evolve nonlinearly they will necessarily form steepened wave fronts, i.e. small-scale shock waves. This is practically unavoidable. And if one attributes any bump or flattened region in the ion-inertial range to KAWs, as is frequently done, the these KAWs are neither linear waves (which would be a completely wrong assertion) nor quasilinearly stabilized waves (which one would not detect because the quasilinear saturation level is miniscule). So the physically reasonable remaining state is that they are small-scale (ion inertial scale) shock waves with all effects which are related to them: ion heating, reflection of electrons, acceleration of electron, excitation of kinetic electron waves like Langmuir, ion-acoustic etc., electron beams along the magnetic field and even first and second harmonic electromagnetic radiation. All kind of dissipative effects in turbulence. this means that the probability of ion-scale shock waves is quite high in a radially expanding (i.e. inhomogeneous) and thus non-stationary solar wind turbulence, processses which have barely yet been considered in formal investigation of solar wind turbulence.

This is to clarify that the probability of the presence of small-scale (ion-inertial scale) shocks superimposed on the turbulent background is relatively high, at least at times.

2) (possible typo) page 5: "particles "loose" their magnetic property" ---> shouldn't be "lose" instead of "loose"?

Thanks, of course.

3) page 9, Figure 2: The region below the solid line "Te/Ti=1" should be generally forbidden" In what sense is "forbidden"? (it could happen anyway, see also remark 3)

Thanks. Deleted.

4) page 9, I have some concerns about the sentence "However, the temperature ratio Te /Ti is variable and usually large, varying between a few and a few tens". And in Fig 2: "In the solar wind the temperature ratio is usually between the solid and dashed lines but mostly closer to the dashed, depending on the exact value of beta_e" (dashed line is Te/Ti=10).
I am not sure about the validity of that statement.

According to
"Newbury, J. A., Russell, C. T., Phillips, J. L., & Gary, S. P. (1998). Electron temperature in the ambient solar wind: Typical properties and a lower bound at 1 AU. Journal of Geophysical Research: Space Physics, 103(A5), 9553–9566. https://doi.org/10.1029/98JA00067"
the ratio Te/Ti tends to be mostly between 4 and 0.5.

And "Wilson III, L. B., Stevens, M. L., Kasper, J. C., Klein, K. G., Maruca, B. A., Bale, S. D., … Salem, C. S. (2018). The Statistical Properties of Solar Wind Temperature Parameters Near 1 au. The Astrophysical Journal Supplement Series, 236(2), 41. https://doi.org/10.3847/1538-4365/aab71c"
found that the typical temperature ratio in the solar wind is actually Te/Ti=1.64 , with a standard deviation of 1.27. So, values even close to Te/Ti=10 are actually rare, and not really a few tens.
Any comment or clarification, please?

Deleted. The two citations included. Thanks for the hint on the two papers.

Well, I did not look into those papers yet. My own experience from measuring electron an ion temperatures in the 80th-90th was that T_e~10 T_i. But this might have been polluted by the location being close to the bow shock as those were the tmeperatures I had access to. I may accept that generally T_e>T_i but strongly doubt the opposite case as I never saw any observation of this kind neither upstream nor downstream of the bow shock. The uncertainty in Wilson et al of being of the same value as the temperature ratio seems to me a statistical effect which I hardly believe because I do not see any efficient bulk cooling mechanism for electrons, and the corona is definitely hotter in the electrons than the ions. There are three cooling mechanisms in a collisionless plasma like the solar wind: radiation of electromagnetic

collisionless plasma like the solar wind: radiation of electromagnetic waves (in a teneous plasma like the solar wind definitely < 1% in energy, because the medium is optically thin), cooling in electron holes (a very interesting process never yet investigated or discussed: this is a two step process: holes are excited by nonlinear trapping of low-energy electrons in the potential of Langmuir or Bernstein modes; this low energy electron component is heated by trapping, part of it escapes and, together with the passing energetic electron component forms an electron beam which has narrow width in momentum and energy, i.e. is cold), finally charge exchange with neutrals which happens in cometary atmospheres and in the upper atmosphere of planets but plays little role in the undisturbed solar wind. It generated hot neutrals and cold electrons (see IBEX observations). Maybe some of the events where old solar wind electrons have been observed mixes in to generated the small $T_e/T_i$ ratios? Otherwise I do ot believe in those observations because the other processes (radiation, electron holes) are probably out in the solar wind, they play a role in the vicinity of shocks or inside shocks, however, though the radiative cooling is rather inefficient. But the cooling by holes is strong!

5) (typo) page 13: "with a factor of proportionality ........ ()1/3". I think the factor (1/3) should be the exponent of the parenthesis (not multiplied as it is now).

Thanks, indeed. Corrected.

6) (possible typo) page 16, Table 1. In the second row, last column, shouldn't be -1/3 instead of -1/6?
(I noticed than the spectral indices ("a-2") are corrected compared to the previous version (I overlooked them before) and also that one table was removed, probably it was not needed. It is also clearer the last column with an explicit b/2 instead of c.)

Thanks again! Yes. True. Typo. Indeed, Table 2 partly doubled this one and was not needed. Instead I replaced it by a few words on advection in the text in order to avoid confusion.

7) page 19: "the power in the second to last expression becomes 1/3" ---> which equation this is referring to? Eq. (43)? If so, which exponent exactly?

Cleared and equation number included.

8) page 27, Caption Figure 3: "power spectrum exhibits a so-called bump at intermediate frequencies of positive slope ~\omega^{1/3}" But later, "The large scatter in the data (weight of line) inhibits distinguishing between K and IK inertial range velocity turbulence" (and similar statements in the main text). So, that sounds like a contradiction, it

should be written in the caption that the fit could be both $\omega^{1/2}$ and $\omega^{1/3}$ (not only the latter).
Similar for Figure 4. "The positive slope $\omega^{1/6}$ in the deformation confirms its origin from pressure balance" I think both slopes 1/6 and 1/4 work as a fit.

True. Distinction is not possible. Expressed in both captions.

9) (typo) page 21, "Sine both evolve" --> "Since both evolve"

Thanks.

10) page 21: "Moreover, from the observations the total $\beta > 1$ though nothing is known about $\beta_i$. We expect $\rho_i > \lambda_i$, and also $\beta_i \sim 1$" This is a contradictory sentence, either beta_i is known or not.

Thanks. Clarified.

11) page 21: please mention what V_0 and U_0 are in the first line of the summary and outlook, that will make this section more self-contained (or at least please make a cross-reference to their definitions after Eq 2 in page 6).

OK. Mentioned in summary. cross-ref is not required then.

Thanks very much for all these very useful comments!